# Obligate sexual reproduction of a homothallic fungus closely related to the *Cryptococcus* pathogenic species complex

Andrew Ryan Passer[1†], Shelly Applen Clancey[1†], Terrance Shea[2], Márcia David-Palma[1], Anna Floyd Averette[1], Teun Boekhout[3,4], Betina M Porcel[5], Minou Nowrousian[6], Christina A Cuomo[2], Sheng Sun[1], Joseph Heitman[1*], Marco A Coelho[1*]

[1]Department of Molecular Genetics and Microbiology, Duke University Medical Center, Durham, United States; [2]Broad Institute of MIT and Harvard, Cambridge, United States; [3]Westerdijk Fungal Biodiversity Institute, Utrecht, Netherlands; [4]Institute of Biodiversity and Ecosystem Dynamics (IBED), University of Amsterdam, Amsterdam, Netherlands; [5]Génomique Métabolique, CNRS, University Evry, Université Paris-Saclay, Evry, France; [6]Lehrstuhl für Molekulare und Zelluläre Botanik, Ruhr-Universität Bochum, Bochum, Germany

*For correspondence:
heitm001@duke.edu (JH);
marco.dias.coelho@duke.edu
(MAC)

†These authors contributed
equally to this work

Competing interest: The authors
declare that no competing
interests exist.

Reviewing Editor: Antonis
Rokas, Vanderbilt University,
United States

**Abstract** Sexual reproduction is a ubiquitous, ancient eukaryotic trait. While most sexual organisms have to find a mating partner, species as diverse as animals, plants, and fungi have evolved the ability to reproduce sexually without requiring another individual. Here, we uncovered the mechanism of self-compatibility (homothallism) in *Cryptococcus depauperatus*, a fungal species closely related to the human fungal pathogens *Cryptococcus neoformans* and *Cryptococcus gattii*. In contrast to *C. neoformans* or *C. gattii*, which grow as a yeast asexually, and produce hyphae, basidia, and infectious spores during sexual reproduction, *C. depauperatus* grows exclusively as hyphae decorated with basidia and abundant spores, thus continuously engaged in sexual reproduction. Through comparative genomics and analyses of mutants defective in key mating/meiosis genes, we demonstrate the *C. depauperatus* sexual cycle involves meiosis and that self-compatibility is orchestrated by an unlinked mating receptor (Ste3**a**) and pheromone ligand (MFα) pair derived from opposite mating types of a heterothallic (self-sterile) ancestor. We identified a putative mating-type (*MAT*) determining region containing genes phylogenetically aligned with *MATa* alleles of other species, and a few *MATα* gene alleles scattered throughout the genome, but no homologs of the mating-type homeodomain genes *SXI1* (*HD1*) and *SXI2* (*HD2*). Comparative analyses suggest a dramatic remodeling of the *MAT* locus possibly owing to reduced selective constraints to maintain mating-type genes in tight linkage, associated with a transition to self-fertility. Our findings support *C. depauperatus* as an obligately sexual, homothallic fungus and provide insight into repeated transitions between sexual reproduction modes that have occurred throughout the fungal kingdom.

## Editor's evaluation

There are various ways in which self-fertility has arisen in the fungal kingdom. This study describes a novel form of self-fertility that evolved in a species closely related to the *Cryptococcus* species causing serious human lung and brain infections, in which sexual development is achieved by self-signaling of a cognate pheromone and pheromone-receptor pair. Through a combination of

high-quality genomic analysis and experimental gene expression and manipulation work, the study significantly adds to our understanding of the evolution and flexibility of fungal breeding systems.

## Introduction

Sexual reproduction, generally defined as the production of viable and fertile offspring by combining genetic information from mating partners of two different types, is a process conserved across the eukaryotic tree of life (*Goodenough and Heitman, 2014*; *Heitman, 2015*). Sexual reproduction has many benefits, such as generating novel genetic combinations and removing deleterious mutations (*Agrawal and Whitlock, 2012*), but it also has many costs, in that it is energetically expensive, time-consuming, and in some species two parents are required to produce one progeny. Therefore, some organisms, such as fungi, balance sexual reproduction with asexual mitotic cycles in order to populate an environmental niche (a strategy known as facultative sexual reproduction) (*Williams, 1975*; *Maynard-Smith, 1978*).

In fungi, sexual reproduction usually involves the fusion of haploid partners of opposite mating type, a situation known as heterothallism. The mechanisms of mating compatibility under heterothallism are quite diverse across fungal taxa and frequently involve genomic structures ranging from a mating-type (*MAT*) locus region containing only one or two genes (e.g., the Mucoromycota and Ascomycota phyla) (*Bennett and Turgeon, 2016*; *Lee and Idnurm, 2017*), to highly complex regions containing several genes at one (bipolar) or two (tetrapolar) *MAT* loci (e.g., most Basidiomycota) (*Coelho et al., 2017*), and even to mating-type chromosomes that exhibit large non-recombining regions resembling sex chromosomes of plants and animals (e.g., *Microbotryum* spp. and *Neurospora tetrasperma*) (*Menkis et al., 2008*; *Branco et al., 2017*; *Sun et al., 2017*; *Branco et al., 2018*). In most of these systems, and similar to the presence of separate sexes in different individuals in animals and plants, the probability of encountering a suitable mating partner when only two mating types exist in a population at equilibrium cannot exceed 50%, which may pose a substantial fitness reduction in environments where population densities are very low and compatible mating partners are scarce (*Hoekstra, 1987*). As a possible evolutionary response to such selective pressures, many fungal species evolved the ability to reproduce sexually without the need for another individual, a state known as homothallism (*Ni et al., 2011*; *Wilson et al., 2015*), analogous to the evolution of hermaphroditism or parthenogenesis in plants and animals (*Jarne and Charlesworth, 1993*; *Neaves and Baumann, 2011*; *Busch and Delph, 2012*). In such fungi, haploid cells are universally compatible for mating, and a single isolate can undergo sexual reproduction alone (*Ni et al., 2011*; *Wilson et al., 2015*). This is usually achieved either by combining the genes of opposite mating types within a single genome or by undergoing mating-type switching (*Ni et al., 2011*; *Gioti et al., 2012*; *Fu et al., 2015*; *Wilson et al., 2015*; *David-Palma et al., 2016*; *Krassowski et al., 2019*; *Cabrita et al., 2021*).

Homothallism in fungi evolved multiple times independently (*Billiard et al., 2012*; *Gioti et al., 2012*; *Wilson et al., 2015*; *Hanson and Wolfe, 2017*; *Krassowski et al., 2019*; *Sun et al., 2019b*; *Cabrita et al., 2021*), indicating that it provides a selective advantage under certain conditions, for instance, (i) to allow reproductive assurance, which may be a substantial benefit in patchy habitats (*Murtagh et al., 2000*; *Nieuwenhuis and Immler, 2016*; *Nieuwenhuis et al., 2018*), (ii) to increase compatibility to promote outcrossing (*Heitman, 2015*), or (iii) even to reduce outbreeding depression by avoiding breaking up locally co-adapted gene complexes (*Epinat and Lenormand, 2009*), which has been hypothesized to be advantageous for pathogenic fungi (*Alby et al., 2009*; *Heitman, 2010*; *Hauser, 2021*).

*Cryptococcus* is a fungal genus within the Basidiomycota that comprises both pathogenic and closely related non-pathogenic saprobic species. The non-pathogenic species currently include *Cryptococcus wingfieldii*, *Cryptococcus amylolentus*, *Cryptococcus floricola*, *Cryptococcus depauperatus*, and *Cryptococcus luteus* (*Liu et al., 2015*; *Passer et al., 2019*). The pathogenic clade, which is responsible for over 200,000 human infections annually (*Rajasingham et al., 2017*), currently has seven recognized species distributed into three subgroups: *Cryptococcus neoformans*, *Cryptococcus deneoformans,* and the *Cryptococcus gattii* species complex (*Hagen et al., 2015*). A new lineage within the *C. gattii* species complex has recently been isolated from middens, midden soil, or tree holes associated with the Southern tree hyrax (*Dendrohyrax arboreus*) in Zambia and termed *C. gattii* VGV; no human infections have thus far been attributed to this novel lineage (*Farrer et al., 2019*).

**eLife digest** Fungi are enigmatic organisms that flourish in soil, on decaying plants, or during infection of animals or plants. Growing in myriad forms, from single-celled yeast to multicellular molds and mushrooms, fungi have also evolved a variety of strategies to reproduce. Normally, fungi reproduce in one of two ways: either they reproduce asexually, with one individual producing a new individual identical to itself, or they reproduce sexually, with two individuals of different 'mating types' contributing to produce a new individual. However, individuals of some species exhibit 'homothallism' or self-fertility: these individuals can produce reproductive cells that are universally compatible, and therefore can reproduce sexually with themselves or with any other cell in the population.

Homothallism has evolved multiple times throughout the fungal kingdom, suggesting it confers advantage when population numbers are low or mates are hard to find. Yet some homothallic fungi been overlooked compared to heterothallic species, whose mating types have been well characterised. Understanding the genetic basis of homothallism and how it evolved in different species can provide insights into pathogenic species that cause fungal disease.

With that in mind, Passer, Clancey et al. explored the genetic basis of homothallism in *Cryptococcus depauperatus*, a close relative of *C. neoformans,* a species that causes fungal infections in humans. A combination of genetic sequencing techniques and experiments were applied to analyse, compare, and manipulate *C. depauperatus'* genome to see how this species evolved self-fertility.

Passer, Clancey et al. showed that *C. depauperatus* evolved the ability to reproduce sexually by itself via a unique evolutionary pathway. The result is a form of homothallism never reported in fungi before. *C. depauperatus* lost some of the genes that control mating in other species of fungi, and acquired genes from the opposing mating types of a heterothallic ancestor to become self-fertile.

Passer, Clancey et al. also found that, unlike other *Cryptococcus* species that switch between asexual and sexual reproduction, *C. depauperatus* grows only as long, branching filaments called hyphae, a sexual form. The species reproduces sexually with itself throughout its life cycle and is unable to produce a yeast (asexual) form, in contrast to other closely related species.

This work offers new insights into how different modes of sexual reproduction have evolved in fungi. It also provides another interesting case of how genome plasticity and evolutionary pressures can produce similar outcomes, homothallism, via different evolutionary paths. Lastly, assembling the complete genome of *C. depauperatus* will foster comparative studies between pathogenic and non-pathogenic *Cryptococcus* species.

The heterothallic reproductive cycle of *C. neoformans* and *C. gattii* (also designated as bisexual mating or opposite-sex mating) has been known since the 1970s and readily occurs under laboratory conditions (*Kwon-Chung, 1975*; *Kwon-Chung, 1976a*; *Kwon-Chung, 1976b*). All of the species in the pathogenic *Cryptococcus* species complex have a bipolar mating system, in which the α and **a** mating types are determined by a single, unusually large (~120kb in size), *MAT* locus that encompasses more than 20 genes (*Lengeler et al., 2002*; *Fraser et al., 2004*; *Loftus et al., 2005*). Among these genes are those encoding the mating-type-specific pheromones (MFα or MF**a**) and G protein-coupled receptors (GPCR) (Ste3α or Ste3**a**) that initiate recognition of compatible mating partners, and the homeodomain transcription factors (HD1/Sxi1α or HD2/Sxi2**a**), which establish cell-type identity and orchestrate progression through the sexual cycle (*Hull et al., 2005*; *Sun et al., 2019a*). In addition, the *MAT* locus contains essential genes (*Fraser et al., 2004*; *Ianiri et al., 2020*) and genes that contribute to virulence (*Sun et al., 2019a*). Importantly, comparative genomic studies with the closely related species *C. amylolentus* uncovered that the single *MAT* locus in the pathogenic *Cryptococcus* species is the result of a fusion of ancestrally unlinked pheromone/receptor (*P/R*) and homeodomain (*HD*) loci, possibly initiated through ectopic inter-centromeric recombination (*Sun et al., 2017*).

Under the proper environmental conditions (e.g., V8 media in dark, dry conditions), *Cryptococcus* cells secrete pheromones unique to the mating type of the cell (**a** cells produce the MF**a** pheromone, and α cells produce the MFα pheromone) (*Davidson et al., 2000*; *McClelland et al., 2002*). These pheromones bind to the Ste3α and Ste3**a** receptors, respectively, which signal through the Ste20 protein to a mitogen-activated protein kinase (MAPK) signaling cascade that includes the Ste11, Ste7, Cpk1, and Ste50 proteins (*Sun et al., 2019a*; *Zhao et al., 2019*). The final target of this signaling

cascade is the transcription factor Mat2 (*Lin et al., 2010*; *Feretzaki and Heitman, 2013*), which directly or indirectly activates genes involved in mating and the yeast-to-hyphal morphological transition, initiated by the *MAT*α parent that extends a conjugation tube towards the enlarged *MAT***a** mating partner (*Zhao et al., 2019*; *Sun et al., 2020*). The two mating partners fuse to form a dikaryotic zygote from which a hyphal filament protrudes and extends to form a dikaryotic hypha. During hyphal growth, fused clamp connections form across the septa to ensure that each hyphal compartment maintains two unfused, paired parental nuclei (*Kwon-Chung, 1976b*; *Lin, 2009*). Following the extension of the hyphal filament, the tip differentiates into a basidium, in which karyogamy and meiosis occur followed by repeated mitotic divisions to produce four chains of spores. Eventually, the spores are released from the basidia, disseminate (acting as infectious propagules) (*Reedy et al., 2007*; *Velagapudi et al., 2009*), and grow as yeast cells until encountering mating stimuli again (*Kwon-Chung, 1975*; *Kwon-Chung, 1976b*; *Sun et al., 2019c*; *Zhao et al., 2019*; *Sun et al., 2020*).

*C. deneoformans* and *C. gattii* can also participate in an unusual form of homothallism, termed unisexual reproduction, during which haploid cells of a single mating type undergo ploidy changes and meiosis to produce genetically identical progeny (*Lin et al., 2005*; *Lin et al., 2010*; *Ni et al., 2011*; *Feretzaki and Heitman, 2013*; *Ni et al., 2013*; *Fu et al., 2015*; *Wilson et al., 2021*). The features of unisexual reproduction are similar to those observed during opposite-sex mating, and there are two ways in which unisexual reproduction is initiated: (i) two cells of the same mating type can fuse, produce hyphae with unfused clamp connections (termed a monokaryon), form basidia, and basidiospores, or (ii) a single cell can undergo endoreplication forming a diploid cell that then produces a similar hyphal filament and completes the sexual cycle (*Lin et al., 2005*; *Zhao et al., 2019*).

Interestingly, many of the species that are closely related to members of the pathogenic *Cryptococcus* species, such as *C. amylolentus*, *C. depauperatus*, and *C. luteus*, are not known to cause disease in plants or animals (*Findley et al., 2009*; *Rodriguez-Carres et al., 2010*) and are instead regarded as saprobes or mycoparasites (i.e., parasites of other fungi) (*Sivakumaran et al., 2003*; *Begerow et al., 2017*). *C. depauperatus*, in particular, was first identified by *Petch, 1931* on scale insects and originally described as the type specimen of *Aspergillus depauperatus*. Although initially considered as an insect-associated fungus, *C. depauperatus* was later reassessed as a possible mycoparasite of the entomopathogenic fungus *Akanthomyces lecanii* considering that (i) specimens from which *C. depauperatus* had been isolated or identified from also contained *A. lecanii* (*Petch, 1931*; *Malloch et al., 1978*; *Samson et al., 1983*; *Kubátová, 1992*), and (ii) the fact that *C. depauperatus*, as with other mycoparasitic basidiomycetes, can produce haustorial branches (*Ginns and Malloch, 2003*). There are only two strains of *C. depauperatus* available: CBS7841, which was isolated from a dead spider in Canada (*Malloch et al., 1978*), and CBS7855, which was isolated from a dead caterpillar in the Czech Republic (*Kubátová, 1992*). Interestingly, *C. depauperatus*, possibly along with *C. luteus*, are the only naturally occurring *Cryptococcus* species with no known yeast phase (*Malloch et al., 1978*; *Kwon-Chung et al., 1995*; *Roberts, 1997*; *Ginns and Bernicchia, 2000*). Indeed, in contrast to the dimorphic growth of *C. neoformans* and *C. gattii*, which are usually yeasts in the asexual stage and produce hyphae, basidia, and basidiospores during the sexual stage, *C. depauperatus* grows by continuously producing hyphae, basidia, and basidiospores under typical laboratory conditions (*Kwon-Chung et al., 1995*) with no budding yeast cells observed during its entire life cycle. This species also displays a slow growth phenotype compared to other *Cryptococcus* species (*Findley et al., 2009*), possibly as a consequence of being continuously engaged in an energetically costly sexual cycle. Early data from fluorescent-activated cell sorting (FACS) indicates the spores are haploid, and random amplified polymorphic DNA (RAPD) analyses found the two strains to be genetically distinct (*Rodriguez-Carres et al., 2010*). Given the geographical range of the two isolates, they may represent different populations or even isolates of two closely related but distinct species. However, given the small sample size (n = 2), it is difficult to know if these differences are strain- or species-specific.

Considering the striking resemblance of its growth with the sexual life cycle of other *Cryptococcus* species, the hypothesis was put forth that *C. depauperatus* is homothallic and only the sexual developmental program is active at any given time during the life cycle of this species (*Malloch et al., 1978*; *Kwon-Chung et al., 1995*; *Rodriguez-Carres et al., 2010*). However, to date, a thorough characterization of the *C. depauperatus* genomes has not been completed, leaving unanswered questions. Here, we carried out an in-depth genomic, genetic, and phenotypic study of *C. depauperatus*. Newly

generated chromosome-level genome assemblies of CBS7841 and CB7855 revealed 98% genome-wide shared identity and a uniform pattern of divergence across the genome. We found a putative *MAT* locus containing genes phylogenetically aligned with *MAT***a** alleles of other species, as well as a few unlinked *MATα* gene alleles, including the *MFα* gene, but no homologs of the mating-type determinants *SXI1* and *SXI2*. This led to the hypothesis that compatible pheromone and pheromone-receptor genes could be the key components underlying the homothallic mating behavior of *C. depauperatus*. *Agrobacterium*-mediated transformation was developed to delete genes by homologous recombination. Deletion of key mating and meiosis genes (*MFα*, *STE3*, and *DMC1*) showed severe defects in basidia and/or spore production, but not in hyphal growth. Furthermore, with the first genetic mutants isolated in this species, we identified recombinant meiotic progeny generated from intra-strain genetic crosses. These data support the hypothesis that *C. depauperatus* is an obligately sexual, homothallic fungal species.

## Results

### CBS7841 and CBS7855 genomes are overall syntenic and present homogeneous genome-wide divergence

To establish the basis of the *C. depauperatus* sexual cycle and explore the genomic variation between the two available isolates, we sequenced, assembled, and annotated the genomes of CBS7841 and CBS7855 with both Oxford Nanopore and Illumina reads. The resulting assemblies are approximately 16.28 Mb (CBS7841) and 16.33 Mb (CBS7855) in size and comprised eight contigs with telomeric repeats $TAA(C)_{4,5}$ at both ends, corresponding to eight chromosomes (*Figure 1*, *Figure 1—figure supplement 1*). Each contig contains a large, open reading frame (ORF)-free region, rich in long terminal repeat (LTR) retrotransposons, which have been shown to be coincident with centromeres in other *Cryptococcus* species (*Janbon et al., 2014*; *Sun et al., 2017*; *Yadav et al., 2018*; *Schotanus and Heitman, 2020*) and are predicted to constitute functional centromeres in *C. depauperatus* (*Figure 1A*).

Gene prediction and annotation identified 6342 and 6329 protein-coding genes, respectively, for CBS7841 and CBS7855. These numbers are in the lower range among the number of genes predicted for other *Cryptococcus* species with complete genomes publicly available (ranging from 6405 in *C. deuterogattii* R254 to 8248 in *C. amylolentus*) (*D'Souza et al., 2011*; *Janbon et al., 2014*; *Farrer et al., 2015*; *Sun et al., 2017*; *Passer et al., 2019*; *Gröhs Ferrareze et al., 2021*). Analysis of Benchmarking Universal Single-Copy Orthologs (BUSCO) revealed a high level of completeness of gene sets, with over 90% of the full-length 4284 BUSCO genes (tremellomycetes_odb10 dataset) being present in both isolates (*Figure 1—figure supplement 1*). This indicates the lower number of protein-coding genes identified in *C. depauperatus* is not the result of systematic gene misannotation, but rather due to genome size contraction. Indeed, the genomes of the two *C. depauperatus* isolates are the smallest among the described *Cryptococcus* species (*D'Souza et al., 2011*; *Janbon et al., 2014*; *Farrer et al., 2015*; *Sun et al., 2017*; *Passer et al., 2019*; *Gröhs Ferrareze et al., 2021*).

According to the current genome assemblies, the chromosome structure seems to be overall conserved between the two *C. depauperatus* isolates, except for five inversions (two large and three small, of which four are coupled with duplicated sequences at the borders; *Figure 1A*, *Figure 1—figure supplement 2*), and the predicted centromeric regions that differ considerably in length between some of the homologous chromosomes (*Figure 1—figure supplement 3*). Electrophoretic karyotypes obtained by pulsed-field gel electrophoresis (PFGE) were completely congruent with the contigs sizes for CBS7841, but revealed a few ambiguities for CBS7855, indicating there might be inherent chromosome instability in this strain (*Figure 1—figure supplement 1C*). The two isolates present ~2% divergence at the nucleotide level, similar to the average intra-lineage genetic divergence observed in species of the *C. gattii* complex (*Farrer et al., 2015*). A sliding window analysis detected a relatively uniform pattern of sequence divergence across the genome and no evidence of introgression between the two isolates as shown by the absence of genomic tracts with nearly zero sequence divergence (except for the rDNA array, composed of 18S-5.8S-28S, which is found as a single unit on Chr 1; *Figure 1—figure supplement 4*). Together, this suggests the two isolates are members of geographically diverging populations that have remained largely isolated since their divergence.

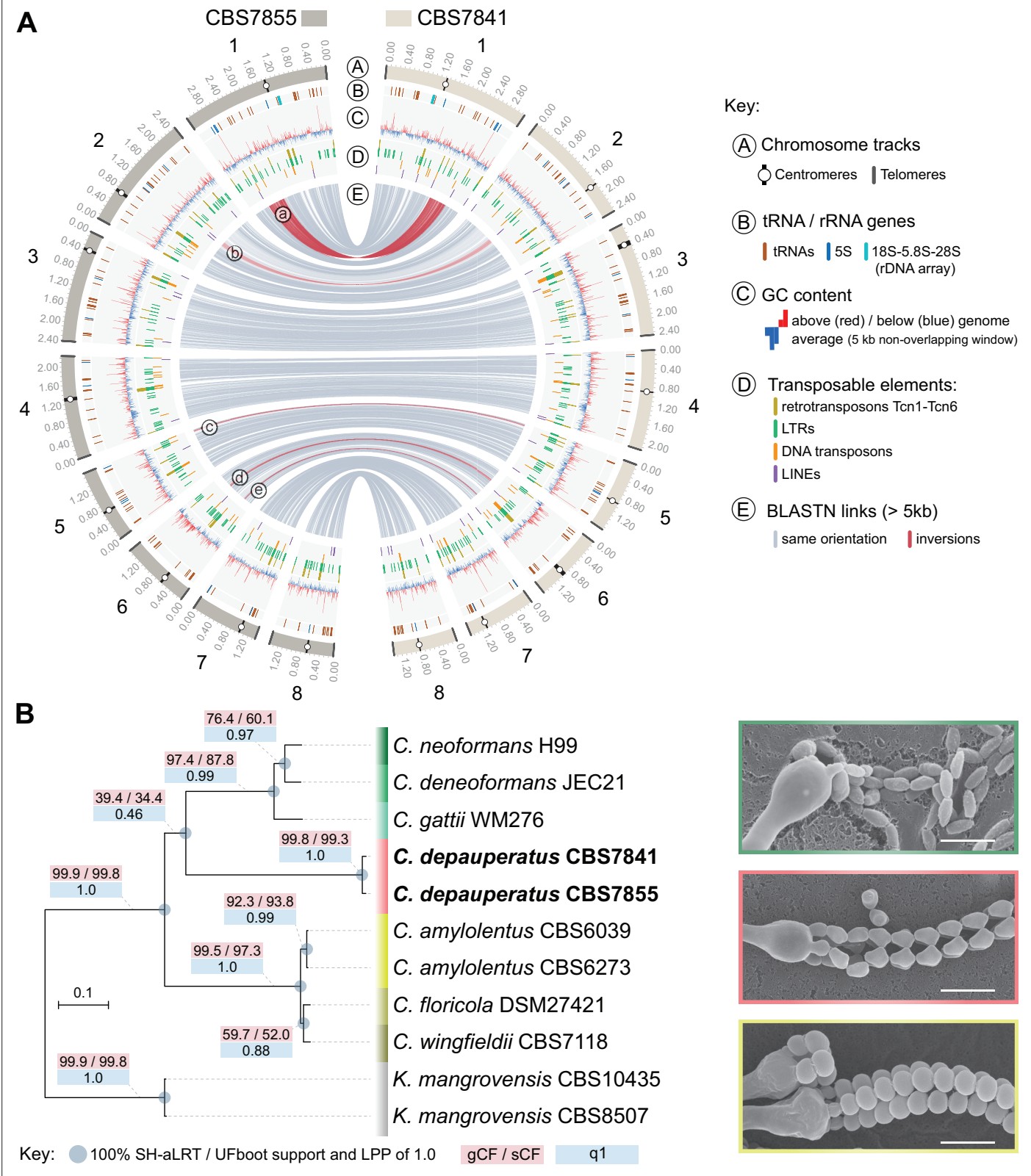

**Figure 1.** Genome-wide comparison between the two *C. depauperatus* strains and phylogenetic placement of *C. depauperatus*. (**A**) Circos plot comparing the genome assemblies of *C. depauperatus* CBS7841 and CBS7855. The two assemblies are overall syntenic, except for five inversions (labelled 'a' to 'e'; see *Figure 1—figure supplement 2* for details). Other genomic features are depicted in different tracks for each chromosome, as shown in the key. (**B**) Genome-based phylogeny recovers *C. depauperatus* as a sister species to the human pathogenic *Cryptococcus* clade. The

*Figure 1 continued on next page*

*Figure 1 continued*

tree was inferred by maximum likelihood using a concatenation-based approach on a data matrix composed of protein alignments of 4074 single-copy genes shared across selected strains of seven *Cryptococcus* species and an outgroup (*Kwoniella mangrovensis*). Log-likelihood of the tree: lnL = −16948764.2158. A coalescence-based tree topology inference obtained by ASTRAL was completely congruent with the concatenation-based phylogeny (see *Figure 1—figure supplement 5*). The reliability of each internal branch was evaluated by 1000 replicates of the Shimodaira–Hasegawa approximate likelihood ratio test (SH-aLRT) and ultrafast bootstrap (UFboot) in the concatenation-based tree, and local posterior probability (LPP) in the coalescence-based tree. Branch lengths are given in number of substitutions per site (scale bar). For each branch of the tree, three additional measures of genealogical concordance are shown: the gene concordance factor (gCF), the site concordance factor (sCF), and quartet support for the main topology (q1)(see 'Materials and methods' for details). Scanning electron microscopy images illustrating sexual reproductive structures (basidia with spore chains) of *C. neoformans* (H99 × KN99; top), *C. depauperatus* CBS7841 (middle) and *C. amylolentus* (CBS6039 × CBS6273; bottom). Scale bars = 5μm.

The online version of this article includes the following source data and figure supplement(s) for figure 1:

**Figure supplement 1.** Statistics of the genome assemblies of CBS7841 and CBS7855 and electrophoretic karyotyping.

**Figure supplement 1—source data 1.** Raw images of clamped homogeneous electrical field (CHEF) gels.

**Figure supplement 2.** Inversions between the two *C. depauperatus* isolates.

**Figure supplement 3.** Centromere length comparison between CBS7841 and CBS7855.

**Figure supplement 4.** Genome-wide divergence (*k*, with Jukes–Cantor correction) of *C. depauperatus* CBS7855 relative to CBS7841 genome reference.

**Figure supplement 5.** Tree topology inferred from the coalescence-based approach.

## Whole-genome phylogenetic analyses place *C. depauperatus* as sister to the human pathogenic *Cryptococcus* clade

To establish evolutionary relationships between *C. depauperatus* and other *Cryptococcus* lineages, we identified 4074 single-copy orthologs shared across selected strains representing main *Cryptococcus* lineages, and *Kwoniella mangrovensis* as an outgroup (*Figure 1B*). Maximum likelihood (ML) phylogenetic inference using both concatenation- and coalescent-based approaches yielded a species phylogeny where all of the internodes received full (100%) support and were recovered consistently in the phylogenies inferred by the two approaches (*Figure 1B*, *Figure 1—figure supplement 5*). In both analyses, *C. depauperatus* appears as sister group to the human pathogenic *Cryptococcus* clade composed of *C. neoformans*, *C. deneoformans*, and the *C. gattii* complex (*Figure 1B*). This phylogenetic placement was, however, inconsistent with previous studies (*Findley et al., 2009*; *Passer et al., 2019*) that provided clade support for the *C. amylolentus* complex as the closest relative of the human pathogenic *Cryptococcus* clade, though these studies employed fewer marker genes. To address these discrepancies, we first measured the genealogical concordance signal by quantifying the proportion of genes (gCF) or sites (sCF) that are concordant with a given branch in the species tree (see 'Materials and methods' for details). This analysis showed that the branch grouping *C. depauperatus* and the human pathogenic *Cryptococcus* species had indeed lower gCF and sCF values compared to other branches in the tree, indicative of some degree of phylogenetic conflict (*Figure 1B*). For example, an sCF of 34.4% for this branch implies that just over a third of the sites informative for this branch supports it. Second, we also examined the tree topologies frequencies supporting each of three competing hypotheses for resolution of each quartet tree obtained from the coalescent-based analysis (*Figure 1—figure supplement 5*). For every quartet tree, there are three possible topologies for how the taxa can be related (noted as T1, T2, and T3 in *Figure 1—figure supplement 5*). In this case, the coalescence-based analysis supported more strongly the main topology (T1), again grouping *C. depauperatus* with the human pathogenic *Cryptococcus* species, compared to the alternative topologies (T2 and T3; *Figure 1—figure supplement 5*). Combined, these analyses suggest that the topology recovered by both concatenation- and coalescent-based approaches represents the best resolution for the *C. depauperatus* placement within the species tree based on current methods and taxa sampling.

## Conserved mating genes are found in linked and unlinked loci in the genome of *C. depauperatus*

To investigate the mechanisms of homothallism in *C. depauperatus*, we first examined the genomes regarding *MAT* gene content and organization. Orthology mapping of genes found within or next to the *MAT* locus in *C. neoformans*, or the *P/R* and *HD* loci in *C. amylolentus* (representing the likely

ancestral state of the genus), identified 25 out of 32 genes analyzed in the genome of *C. depauperatus* (*Figure 2—source data 1*). Over half of these genes (n = 15) were confined to a region spanning ~190 kb on Chr 4 in both CBS7841 and CBS7855, which we designated as the putative *MAT* locus (*Figure 2A*, *Figure 2—source data 1*). Genes in this region with predicted key functions during mating include the pheromone receptor (*STE3*) and the p21-activated kinase (*STE20*) genes. The remaining 10 queried genes were found scattered throughout the genome, including *STE11*, *RUM1*, and *BSP1* on Chr 7, *STE12* and *ETF1* on Chr 1, and a single mating pheromone (*MF*) precursor gene on Chr 6 (*Figure 2A*, *Figure 2—source data 1*) predicted to encode a 37-amino acid product with a C-terminal CAAX motif characteristic of fungal mating pheromones (*Kües et al., 2011*; *Coelho et al., 2017*; *Figure 2B*). Among the genes missing in *C. depauperatus*, it was surprising to note the complete absence of homologs of the homeodomain transcription factors *SXI1* (*HD1*) and *SXI2* (*HD2*) as these genes invariably have central roles in mating-type determination and regulation of the sexual cycle in other basidiomycetes (reviewed in *Coelho et al., 2017*). To validate this further, we performed domain-based searches for proteins containing homeodomains (InterPro entries: IPR008422 and IPR001356). This analysis yielded five significant candidate genes (IDs in CBS7841: L203_102494, L203_103760, L203_103764, L203_104414 and L203_104715), each with a recognizable ortholog in the *C. neoformans* genome: *HOB2* (CNAG_01858), *HOB7* (CNAG_05176), *HOB6* (CNAG_05093), *HOB3* (CNAG_06921), and *HOB4* (CNAG_04586). A previous study in *C. neoformans* reported that these genes are general stress responsive transcription factors (*Jung et al., 2015*), and we presume they might have similar roles in *C. depauperatus*, thus different from the homeodomain transcription factors Sxi1 and Sxi2. Hence, *C. depauperatus* has lost both the *SXI1* and the *SXI2* genes representing, to our knowledge, a unique case in the Basidiomycota.

The identification of *MF* and *STE3* genes unlinked in the genome defined a second distinctive feature of *C. depauperatus* (*Figure 2A*). Sequence alignments and phylogenetic analyses further showed that the predicted product of the *MF* gene is highly similar to other α/A1 pheromones (*Figure 2B*), whereas Ste3 clusters together with **a**/A2 alleles from other *Cryptococcus* species (*Figure 2C*), indicating that they might constitute a compatible pheromone-receptor pair. This suggests a model where homothallism in *C. depauperatus* could have evolved through the combination of two mating types within the same haploid genome. By analyzing individual genealogies of other shared genes found in the *C. neoformans MAT* locus, we identified four additional genes (*MYO2*, *STE12*, *STE11*, and *STE20*) that displayed mating type-specific signatures (*Figure 2—figure supplement 1*). The *MYO2* gene residing at the putative *MAT* locus of *C. depauperatus* grouped together with **a**/A2 alleles from other *Cryptococcus* species, representing the second instance (along with *STE3*) of a mating type **a**/A2 allele present at *MAT*. In contrast, the *C. depauperatus STE11* and *STE12* genes, which are both encoded outside the putative *MAT* locus, clustered together with other α/A1 alleles. Intriguingly, the *C. depauperatus STE20* gene, although clustering more closely with α/A1 alleles, resides at the predicted *MAT* of *C. depauperatus*. One explanation could be that a recombination event, in the form of gene conversion, replaced *STE20***a** with the corresponding α allele. Such gene conversion events were previously shown to occur in the *MAT* locus of *C. neoformans* (*Sun et al., 2012*). Interestingly, these four genes displayed different phylogenetic histories. The *STE12*, *MYO2*, and *STE20* genes exhibited trans-specific polymorphism across all the *Cryptococcus* lineages, with alleles associated with the α/A1 mating type of all species branching together rather than each clustering with the **a**/A2 allele from the same species. Conversely, *STE11* displayed trans-specific polymorphism only across the *Cryptococcus* pathogenic clade (*Figure 2—figure supplement 1*). This pattern suggests that *STE12*, *MYO2*, and *STE20* became linked to *MAT* before the diversification of *Cryptococcus*, whereas *STE11* was integrated into *MAT* more recently, possibly coinciding with the expansion of *MAT* in the common ancestor of the human pathogenic *Cryptococcus* clade (*Fraser et al., 2004*).

## The *C. depauperatus MAT* locus contains genes associated with both *HD* and *P/R* loci of *C. amylolentus* suggestive of a past fusion event between the two regions

Given the phylogenetic placement of *C. depauperatus*, and the fact that many of the mating genes were found in unlinked loci in the genome, we wondered if the *C. depauperatus MAT* locus structure would resemble either that of the tetrapolar species *C. amylolentus* (*Sun et al., 2017*; *Passer et al., 2019*), with *P/R* and *HD* loci unlinked on separate chromosomes representing a more ancestral

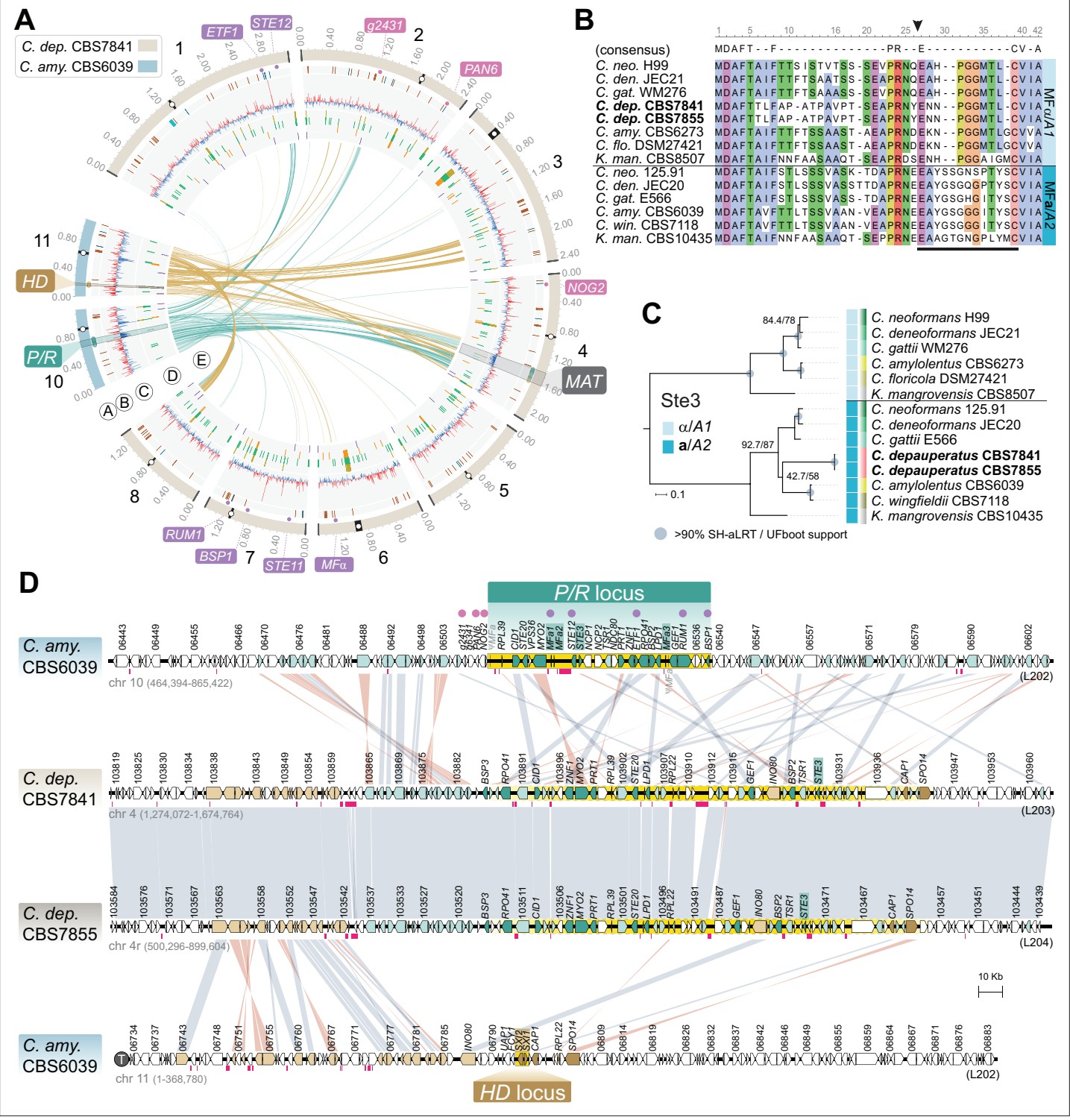

**Figure 2.** The predicted *MAT* locus of *C. depauperatus*. (**A**) Circos plot depicting the distribution of BLASTN hits between the *C. amylolentus* chromosomes containing the *P/R* (Chr 10; links colored in teal blue) and *HD* (Chr 11; links colored in gold) *MAT* loci and the *C. depauperatus* CBS7841 chromosomes. Genes residing within the *MAT* locus of *C. neoformans* whose orthologs in *C. depauperatus* are predominantly clustered on Chr 4 (putative *MAT*) are depicted as small circles colored in teal blue and gold next to the chromosome tracks. Other small circles illustrate genes within (purple) or bordering (pink) the *C. neoformans MAT* locus that are found dispersed in the *C. depauperatus* genome (see *Figure 2—source data 1*). Other genomic features are as given in the legend of *Figure 1A*. (**B**) Sequence alignment of MFα/A1 and MFa/A2 pheromone precursors showing that the *C. depauperatus* mature pheromone is highly similar to MFα/A1 pheromones of other *Cryptococcus* species. The black arrowhead denotes

*Figure 2 continued on next page*

*Figure 2 continued*

the predicted cleavage site, giving rise to the peptide moiety of the mature pheromone (indicated by a black bar). (**C**) Maximum likelihood phylogeny of Ste3α(A1)and Ste3a(A2)pheromone receptors from different *Cryptococcus* species and *Kwoniella mangrovensis* (outgroup). *C. depauperatus* Ste3 sequences cluster together with a/A2alleles from other species. Internal branch support was assessed by 10,000 replicates of Shimodaira–Hasegawa approximate likelihood ratio test (SH-aLRT) and ultrafast bootstrap (UFboot) and are depicted by gray circles when well supported (>90%). Branch lengths are given in number of substitutions per site. (**D**) Synteny maps of the *C. amylolentus* CBS6039 *P/R* (top panel) and *HD* (bottom panel) *MAT* loci compared to the predicted *MAT* region of *C. depauperatus* CBS7841 and CBS7855. The regions spanning the proposed *HD* and *P/R* loci in *C. amylolentus* are highlighted in yellow. Genes at the predicted *MAT* region of *C. depauperatus* with a corresponding ortholog in *C. amylolentus* found within or flanking the *P/R* locus are colored, respectively, in dark or light teal blue. Genes flanking the *HD* locus in *C. amylolentus* with a corresponding ortholog in *C. depauperatus* at the predicted *MAT* region are shown in light gold, while those found within the fused *MAT* locus of *C. neoformans* are colored in dark gold. No homologs of the homeodomain transcription factor genes *HD1* (*SXI1*) and *HD2* (*SXI2*) were detected in the genome of *C. depauperatus*. Vertical gray or pink bars connect orthologs with the same or inverted orientation. Pink horizontal bars below the genes represent repeat-rich regions containing transposable elements or their remnants.  ϕ *MFa* indicates pheromone gene remnants in CBS6039, and the telomere of Chr 11 in *C. amylolentus* is depicted by a circled 'T'.

The online version of this article includes the following source data and figure supplement(s) for figure 2:

**Source data 1.** Genes within or adjacent to the *C. neoformans* H99 *MAT* locus or the *P/R* and *HD MAT* loci of *C. amylolentus* CBS6039, and corresponding orthologs in *C. depauperatus*.

**Figure supplement 1.** Individual genealogies of genes within the *MAT* locus of *C. neoformans* exhibiting different level of trans-specific polymorphism.

**Figure supplement 2.** Genome comparison showing the distribution of BLASTN hits (>0.5kb) represented as links between Chr 4 of *C. depauperatus*, on which the predicted *MAT* locus is located, and all 14 chromosomes of (**A**) *C. amylolentus* CBS6039 and (**B**) *C. neoformans* H99.

**Figure supplement 3.** The *MAT* region of *C. depauperatus* contains genes that are flanking the centromeres of *MAT*-containing chromosomes in *C. neoformans* and *C. amylolentus*.

state, or mirror instead the derived bipolar configuration of *C. neoformans* where these two loci are genetically linked (***Lengeler et al., 2002***; ***Fraser et al., 2004***). Whole-genome comparisons between *C. amylolentus* and *C. depauperatus* revealed that many regions of the *C. amylolentus* *P/R*- and *HD*-containing chromosomes (Chr 10 and Chr 11, respectively) are partially syntenic to Chr 4 of *C. depauperatus*, on which the *MAT* locus resides (***Figure 2A***, ***Figure 2—figure supplement 2A***).

Detailed synteny maps further showed that several genes within or adjacent to the *C. amylolentus* *HD* and *P/R* loci are juxtaposed in *C. depauperatus* (***Figure 2D***), suggesting the two regions fused during evolution, presumably via chromosomal translocation. Through a similar comparison, we additionally found that most of the length of *C. depauperatus* Chr 4 corresponds to genomic segments of Chrs 1 and 5 of *C. neoformans* (***Figure 2—figure supplements 2B and 3A***). Importantly, an ~500 kb region on *C. depauperatus* Chr 4 surrounding the putative *MAT* locus largely combines two segments of *C. neoformans* Chr 5: one, more telomere-proximal, includes *MAT* and its neighboring regions; the other is centromere-proximal and contains some of the genes flanking *C. neoformans* CEN5 (***Figure 2—figure supplement 3***). It seems, however, the corresponding centromere in *C. depauperatus* was lost during evolution, possibly associated with gross chromosomal rearrangements. Other smaller segments of *C. depauperatus* Chr 4 correspond to regions of *C. amylolentus* Chrs 3, 4, 5, 6, 7, and 12 or Chrs 4, 6, 7, and 13 of *C. neoformans* (***Figure 2—figure supplement 2***), underscoring that a vast number of chromosomal rearrangements have occurred in these species since their last common ancestor.

Taken together, these findings indicate the putative *MAT* locus of *C. depauperatus* structurally resembles the derived bipolar *MAT* locus of *C. neoformans* but evolved independently and underwent substantial remodeling via genome reshuffling and gene loss.

## Key mating and meiosis genes are present in *C. depauperatus* and upregulated during sporulation-inducing conditions

Meiosis in *Cryptococcus* occurs following karyogamy or endoreplication in basidia formed at the apexes of aerial hyphae to reduce the DNA content by half before sporulation. The genes that support a meiotic pathway, usually termed the 'meiotic toolkit,' are highly conserved from yeasts to humans (***Schurko and Logsdon, 2008***). Of the 30 genes defined as 'core' meiotic genes, all except *MSH4* and *MHS5* were unequivocally found in the *C. depauperatus* genomes (***Figure 3—source data 1***). The absence of Msh4 and Msh5 orthologs was confirmed by TBLASTN searches employing the *C. neoformans* genes as query sequences. Whereas this procedure failed to return orthologs in *C.*

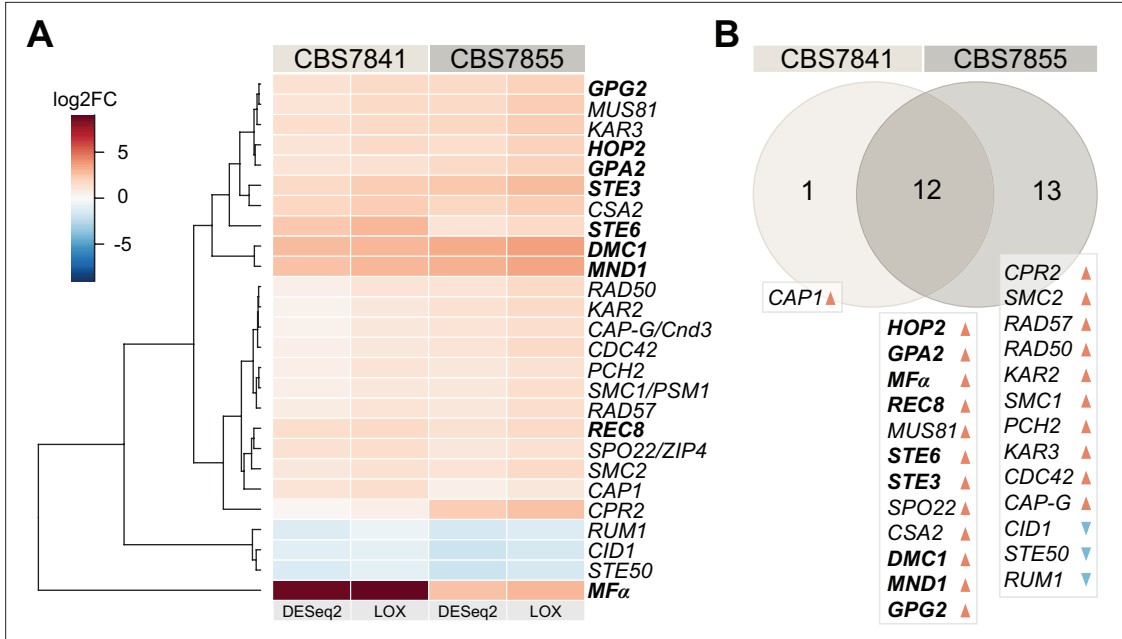

**Figure 3.** *C. depauperatus* displays upregulation of key mating and meiotic genes in sporulating conditions. (**A**) Heatmap of gene expression analysis of mating and meiosis-related genes for CBS7841 and CBS7855 (see *Figure 3—source data 1* and *Figure 3—source data 2* for the complete list of genes). Log$_2$ fold changes (log$_2$FC) for conditions conducive for sporulation (solid medium) vs. conditions that inhibited sporulation (liquid media) were determined with DESeq2 and LOX algorithms. Genes shown are differentially expressed in at least one of the two strains using the following thresholds for differential expression: log$_2$FC ≥ ±1, false discovery rate (FDR) (p-adj, DESeq2) ≤ 0.05, p-value=1 (LOX) in the corresponding direction. RNA-seq was performed in triplicate for each strain and condition, and the heatmap shows the mean values across samples. Clustering and generation of heatmaps was done in R (v3.5.1). (**B**) Venn diagram showing shared and unique differentially expressed genes (DEGs) between the two strains relative to RNA-seq conditions reported in panel (**A**). Upregulated and downregulated genes are indicated, respectively, by orange and blue arrowheads, and key mating and meiotic DEGs in both strains are depicted in boldface.

The online version of this article includes the following source data for figure 3:

**Source data 1.** Mating and meiosis genes and their expression under sporulation (solid) vs. non-sporulation (liquid) conditions.

**Source data 2.** *C. depauperatus* gene expression analysis comparing sporulation (solid) vs. non-sporulation (liquid) conditions.

*depauperatus*, it could recover orthologs in other *Cryptococcus* species and detect paralogous hits in the *C. depauperatus* genomes (e.g., Msh2 and Msh6). Msh4 and Msh5 are components of the major crossover formation pathway in *Saccharomyces cerevisiae* coordinating crossing over with formation of the synaptonemal complex and are required for resolution of Holliday junctions, while two other partners, Mus81-Mms4/Eme1, participate in the minor pathway of crossover resolution (de los *de los Santos et al., 2003*; *Argueso et al., 2004*; *Zakharyevich et al., 2012*). Msh4-Msh5, but not Mus81-Mms4/Eme1, promotes crossovers that display interference. Therefore, the loss of Msh4-Msh5 in *C. depauperatus* suggests that some of the mechanisms of chromosome pairing, crossover formation, and resolution might be accomplished through a reduced machinery, although evidence for synaptonemal complexes in the basidia was provided by transmission electron microscopy in a previous study (*Kwon-Chung et al., 1995*).

Whole-genome sequencing of *C. depauperatus* revealed that key genes involved in mating-type determination and sexual development in other *Cryptococcus* species are not all found in a single *MAT* locus. As a result, we hypothesized that some of these genes could be under alternative regulation for gene expression, particularly those involved in karyogamy, mating, and meiosis. To determine whether we could observe activation of mating and meiosis genes in their native environment, we compared the transcriptomic profile of CBS7841 and CBS7855 grown under conditions that stimulate sporulation (growth on solid medium) and conditions that inhibit sporulation (growth in a liquid culture). RNA-seq analysis revealed 26 genes with a role in mating and meiosis (out of 95 genes evaluated; *Figure 3—source data 1*) that were differentially expressed under the conditions tested in at least one of the two strains, for log$_2$ fold change (FC) ≥±1 and a false discovery rate (FDR) ≤ 0.05 in

DESeq2, and *P*-value of 1 (in the corresponding direction) in LOX analyses (*Figure 3A*, *Figure 3—source data 1*). Among these genes, 12 are shared by the two strains and upregulated in conditions of sporulation (*Figure 3B*). Particularly striking was the upregulation of *MFα*, which had the highest expression level in CBS7841 ($\log_2$ FC = 8.69 in DESeq2), and the significantly higher expression of genes involved in the pheromone response pathway (*GPA2*, *GPG2*, and *STE3*) and *STE6*, a gene that encodes the transporter for mature **a**- and α-pheromones (*Figure 3*; *Hsueh and Shen, 2005*). *DMC1*, which encodes a meiotic recombination protein that plays a central role in homologous recombination during meiosis (*Bishop et al., 1992*), was also upregulated during sporulation, as well as *MND1* and *HOP2* whose products form a complex to ensure proper chromosome paring and nuclear division (*Tsubouchi and Roeder, 2002*), and Rec8 that mediates cohesion between sister chromatids (*Buonomo et al., 2000*). CBS7855 also displayed upregulation of *CPR2*, a pheromone receptor-like gene that elicits unisexual reproduction in *C. deneoformans* when the corresponding ortholog is over-expressed (*Hsueh et al., 2009*). It is important to note that the observed expression differences between the two strains under the same experimental conditions may be due to intrinsic variability or result from different proportions of sporulating cells at the end of the incubation period prior to isolation of RNA. Overall, these findings suggest that the mating and meiotic pathways are functional and activated during sporulation in *C. depauperatus*.

## Ectopic expression of *C. depauperatus MFα* pheromone in *C. neoformans* induces hyphal formation

Our finding that *MFα* and *STE3***a** coexist in the same haploid genome led us to hypothesize that the *C. depauperatus* sexual cycle might be initiated by an autocrine signaling loop wherein cells produce α pheromone that then binds to its cognate receptor (Ste3**a**) activating the downstream signaling cascades required for sexual development. To experimentally determine whether ectopic expression of *C. depauperatus MFα* gene could elicit a mating response in a strain that contains the receptor for *MFα* (*STE3***a**), the *MFα* gene was cloned from *C. depauperatus* and introduced into the safe haven locus of *C. neoformans MAT***a** and *MATα crg1Δ* mutant strains (KN99**a** *crg1Δ* and H99α *crg1Δ*, respectively) (*Arras et al., 2015*). Crg1 is a regulator of G protein signaling that negatively regulates Gα proteins activated by the pheromone receptors (by stimulating GTP hydrolysis by Gα-GTP) and downregulates the signal from a pheromone-bound Ste3 receptor (*Nielsen et al., 2003*; *Wang et al., 2004*). Thus, *crg1Δ* mutants have been shown to be hypersensitive to mating stimuli (*Fraser et al., 2003*; *Wang et al., 2004*) and, therefore, can serve as a facile assay to detect aspects of mating. As predicted, expression of the *C. depauperatus MFα* pheromone gene (*MFα*$_{Cd}$) in the KN99**a** *crg1Δ* strain, but not in H99α *crg1Δ*, resulted in robust hyphal development on filament agar after 14 days of incubation (*Figure 4A*). This is functional evidence confirming that the *C. depauperatus* pheromone gene is of the α variety and that expression of the *MFα* gene outside of the *MAT* locus can induce filamentation.

If the hyphal growth observed in the *C. neoformans* strain expressing *MFα*$_{Cd}$ is the result of a constitutively active pheromone response pathway, then we expected that deletion of the *C. neoformans STE3***a** receptor should disrupt the heterologous pheromone response and inhibit hyphal growth. Indeed, deletion of the *STE3***a** gene in KN99**a** *crg1Δ MFα*$_{Cd}$ background completely abolished hyphal development (*Figure 4B*). Likewise, if the pheromone could no longer be exported from the cell, the positive feedback loop would be disrupted, which would similarly prevent hyphal growth. Consistent with this hypothesis, filamentation was no longer observed after deleting the pheromone exporter *STE6* in the KN99**a** *crg1Δ MFα*$_{Cd}$ strain (*Figure 4B*). Although the *C. neoformans* and *C. depauperatus* mature MFα protein sequences differ by two amino acids (*Figure 2C*), these results indicate that the *C. depauperatus* MFα pheromone can apparently undergo the same post-translational modifications and utilize the same machinery as the native *C. neoformans* pheromone to induce a mating response.

## Mutants defective in critical mating and meiosis pathway components disrupt normal basidia formation and sporulation but not hyphal growth in *C. depauperatus*

In light of these results, we deemed likely that MFα pheromone production in *C. depauperatus* may itself activate an autocrine signaling response via activation of the endogenous Ste3**a** pheromone receptor. To provide direct evidence for this hypothesis, we sought to delete key genes in *C.*

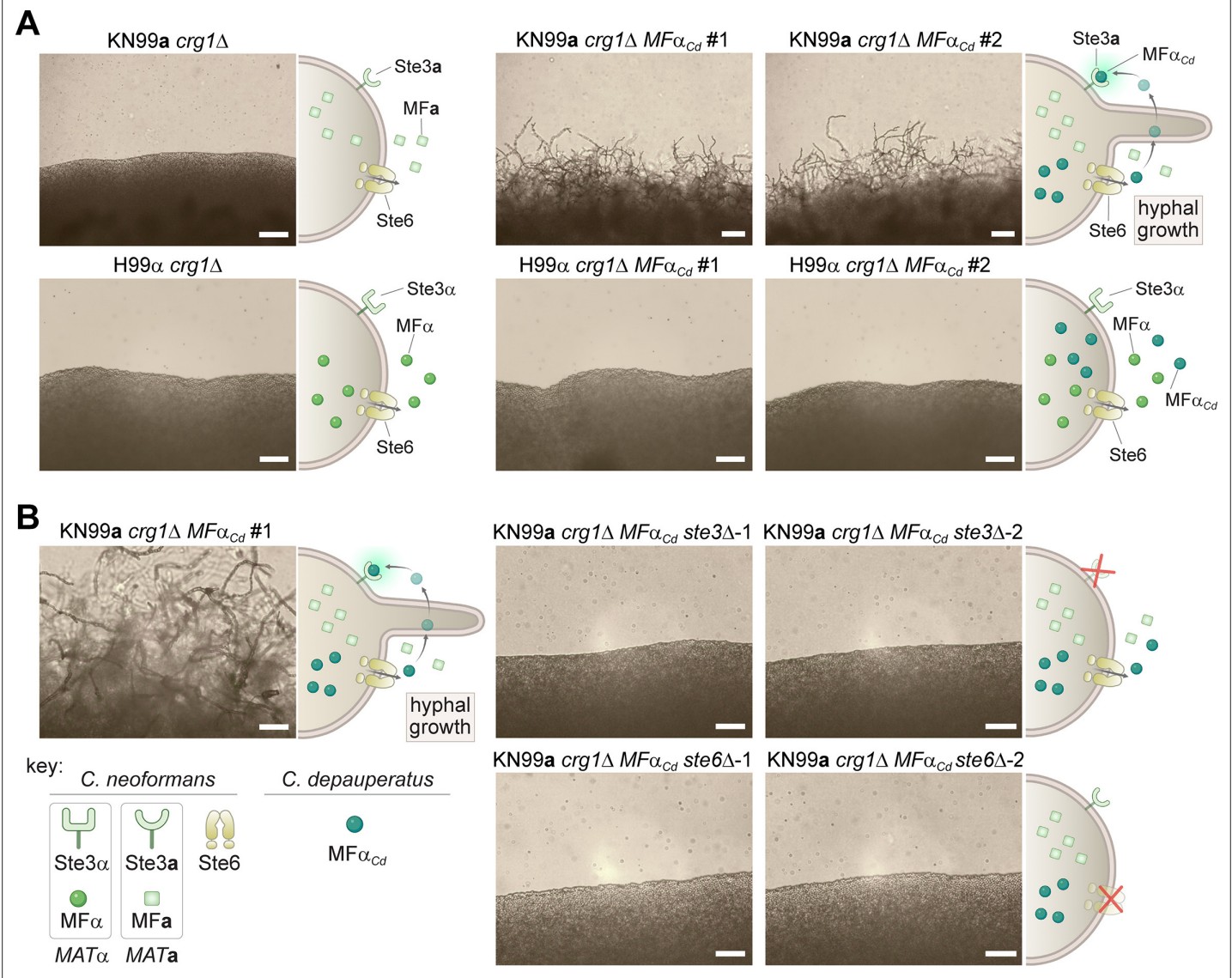

**Figure 4.** Ectopic expression of the *C. depauperatus MFα* gene in *C. neoformans* induces self-filamentation through Ste3 and Ste6. (**A**) The *C. depauperatus MFα* gene was introduced ectopically into *C. neoformans MATα* (H99α *crg1Δ*) and *MATa*(KN99a *crg1Δ*) strains, and transformants harboring the transgene were assessed for their ability to filament. Light microscopy images of cell patches on filament agar after 6days of incubation at room temperature in the dark. Scale bars represent 50μm for all images except for the KN99a *MFα_Cd* transformants, where it denotes 100μm. (**B**) *STE3* and *STE6* deletion mutants were constructed in the modified KN99a *crg1Δ MFα_Cd* self-filamentous strain. After incubation on filament agar, no filamentation was observed in the *ste3Δ* and *ste6Δ* mutants, indicating that both proteins are required for *MFα_Cd* to stimulate self-filamentation in *C. neoformans*. All scale bars represent 50μm.

The online version of this article includes the following source data for figure 4:

**Source data 1.** Raw images of gels validating *C. neoformans* WT, *ste3Δ* and *ste6Δ* strains expressing the *C. depauperatus* pheromone.

*depauperatus* involved in the pheromone response pathway (*MFα* and *STE3a*) and meiosis (*DMC1*), which had previously shown strong upregulation under sporulation conditions (***Figure 3***). However, few in-depth genetic studies had been performed in *C. depauperatus,* so there was no established transformation system for this species. We tested multiple transformation approaches (including biolistic transformation and electroporation), but only *Agrobacterium tumefaciens*-mediated transformation (ATMT) proved to be capable of delivering plasmids conferring drug resistance into *C. depauperatus.* Successful transformation of *C. depauperatus* also required constructing drug-resistance cassettes, containing a drug-resistance gene optimized for use in *Cryptococcus*, flanked by the *C. depauperatus*

actin (*ACT1*) promoter and the phosphoribosyl anthranilate isomerase (*TRP1*) terminator sequences (see 'Materials and methods' for details).

By employing ATMT in *C. depauperatus*, we first isolated transformants with ectopic integration of the gene conferring resistant to nourseothricin (*NAT*) (*Figure 5—figure supplement 1*). Notably, *C. depauperatus* was highly resistant to concentrations of nourseothricin that are typically used for *Cryptococcus* (100 µg/mL) and required three times as much drug (300 µg/mL) to observe growth inhibition. ATMT was subsequently applied to delete the *MFα*, *STE3a*, and *DMC1* genes. We obtained a single deletion mutant for *MFα*, two independent *STE3* deletion mutants, and one *DMC1* deletion mutant, all in the CBS7841 background.

Following PCR confirmation of gene deletion, mutants were analyzed by light microscopy and scanning electron microscopy for phenotypic defects. *C. depauperatus mfαΔ* and *ste3aΔ* mutants displayed strikingly similar hyphal morphology, and defects in basidia maturation compared to wild-type CBS7841 (*Figure 5*). Notably, the two mutants showed a significant reduction in the frequency of basidia with spore chains (*Figure 5B*, *Figure 5—source data 1*) and exhibited unsporulated basidia in both actively growing and older hyphae that were significantly smaller compared to wild-type unsporulated basidia (*Figure 5C*, *Figure 5—figure supplement 2*, *Figure 5—source data 2*). Besides the defects in basidia maturation, apical branching (tip-slitting) was observed near the termini of single hyphal filaments in both mutants (*Figure 5A*); a phenotype that resembles that of *Neurospora crassa* actin mutants (*Virag and Griffiths, 2004*) and may as well reflect a defect in hyphal polarity. An *mfαΔ ste3Δ* double mutant isolated from progeny of *mfαΔ* x *ste3Δ* co-cultures (see next section) displayed similar defects as the single mutants (*Figure 5B*). Despite this, none of the mutants had complete impairment of hyphal formation, indicating that this process in *C. depauperatus* can occur independently of pheromone-receptor signaling.

Interestingly, a low percentage of the basidial population in the *mfαΔ* and *ste3Δ* mutants (<5.5% in the *mfαΔ* and <4.5% in the *ste3Δ*; *Figure 5B and C*) could, however, reach maturation and undergo sporulation. In such cases, the basidia diameter was not significantly different from wild-type sporulating basidia (*Figure 5B and C*). This residual sporulation could be the result of various compensatory processes, to be explored in future studies, including (i) constitutive basal activity of *STE3a* in the *mfαΔ* mutant; (ii) transgressive activation in the *ste3aΔ* mutant of other Ste3-like receptors such as Cpr2, which is known in *C. deneoformans* to compete with the Ste3 receptor for signaling and whose overexpression elicits unisexual reproduction (*Hsueh et al., 2009*); or (iii) other events such as aneuploidy. Together, our results indicate that disruption of *STE3* or *MFα* severely affects basidial maturation in *C. depauperatus* and, consequently, has a strong impact in spore production.

Similarly, a complete sporulation defect phenotype was observed in the absence of *DMC1* (*Figure 5*), although the *dmc1Δ* mutant produced hyphal structures with similar appearance to the wild-type, in contrast to the tip-splitting hyphal phenotypes of the pheromone and pheromone-receptor mutant strains. Additionally, the basidial diameter of the *dmc1Δ* mutant was not significantly different from wild-type unsporulated basidia (i.e., basidia observed in younger hyphae prior to sporulation; *Figure 5C*), indicating that the disruption of this meiotic essential gene does not impair basidial differentiation and maturation, similar to the findings previously reported for *C. deneoformans* (*Liu et al., 2018*). Overall, these findings provide additional evidence that *C. depauperatus* sexual reproduction is mediated via autocrine pheromone-receptor signaling and involves a meiotic cycle.

## Exogenous pheromone stimulates sporulation of a *C. depauperatus* mutant lacking pheromone

We next sought to answer whether the phenotype of the *mfαΔ* pheromone-less mutant could be rescued when supplied with exogenous pheromone. To test this, we first attempted to stimulate sporulation in the *mfαΔ* mutant using a confrontation assay where wild-type and mutant strains were placed close together on a mating plate, but without contacting each other. However, we did not to observe any basidia with spores in the mutants across the gap in the confrontation assays (*Figure 6—figure supplement 1*). We hypothesized that this could be due to low diffusion and concentration of the lipid-modified mating pheromone and/or insufficient proximity between interacting cells. To circumvent this, we next co-cultured the *mfαΔ* strain with the *ste3Δ* deletion mutant (which still produces and secretes the mature α pheromone) and examined if the number of basidia with spore chains significantly increased compared to the residual numbers observed when both mutants were

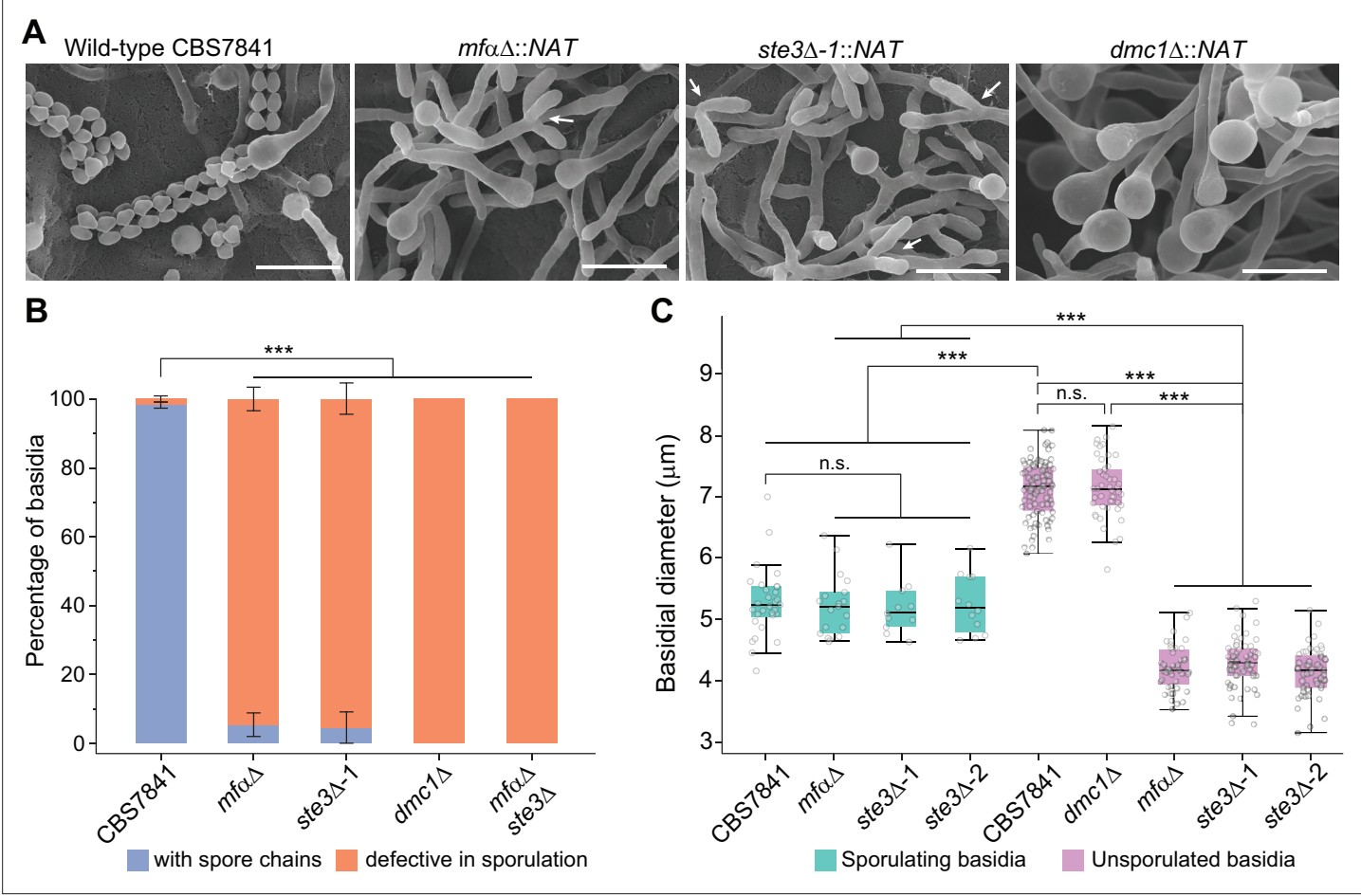

**Figure 5.** *C. depauperatus mfα∆* and *ste3∆* mutants display defects in basidia maturation and sporulation, and the *dmc1∆* mutant shows impaired sporulation despite achieving basidial maturation. (**A**) Scanning electron microscopy images of wild-type CBS7841, *mfα∆* (SEC831), *ste3∆* (SEC836), and *dmc1∆* (SEC866) deletion mutants. Cells were imaged following 1week of incubation on V8 medium at room temperature in the dark. Images were taken at ×3500 magnification; bars = 10µm. Arrows show examples of the tip splitting phenotype observed in *mfα∆* and *ste3∆* mutants. (**B**) Quantification of basidia-producing spores and basidia defective in sporulation. Strains were incubated on Murashige–Skoog (MS) medium for 10days. Up to 100 basidia were evaluated in each case across at least three independent images, and the percentage of basidia with spores (blue) and those defective in sporulation (i.e., bald basidia; orange) is represented. Single mutants (*mfα∆*, *ste3∆*, and *dmc1∆*) were constructed via *A. tumefaciens*-mediated transformation, and the *mfα∆ ste3∆* double mutant was isolated from progeny of *mfα∆* × *ste3∆* co-cultures (basidia #6 isolate number 1). Error bars represent standard error of the mean (see *Figure 5—source data 1*). (**C**) Box and whisker plots showing the diameter of sporulating and unsporulated basidia in wild-type and mutant strains. Strains were incubated on MS medium for 25days, imaged at ×12.5 magnification, and the basidial diameter was measured with ImageJ software (see *Figure 5—figure supplement 2*). Shaded boxes and black line represent the interquartile ranges and median value, respectively. Outliers are included (see *Figure 5—source data 2*). Statistical significance in panels (**B**) and (**C**) was determined with a one-way ANOVA and Tukey's post hoc test. *** Significant at p<0.0001; n.s., not significant.

The online version of this article includes the following source data and figure supplement(s) for figure 5:

**Source data 1.** Frequency of basidia defective in sporulation (bald basidia) and basidia with spores in *C. depauperatus* wild-type (CBS7841) and mutant strains, and one-way ANOVA and Tukey's HSD post hoc statistical tests for frequencies of bald basidia.

**Source data 2.** Diameter of unsporulated and sporulating basidia in *C. depauperatus* wild-type (CBS7841) and mutant strains (*mfα∆*, *ste3∆*, and *dmc1∆*) following incubation on Murashige–Skoog (MS) medium for 25days at room temperature in the dark.

**Figure supplement 1.** Schematic of the transformation system optimized for *C. depauperatus* and genotypic analysis of *C. depauperatus* transformants.

**Figure supplement 1—source data 1.** Raw images of gels validating the *C. depauperatus* deletion mutant strains.

**Figure supplement 2.** Representative microscopy images of (**A**) wild-type CBS7841, (**B**) *dmc1∆* (SEC866), *mfα∆* (SEC831), (**C**) *ste3∆-1* (SEC836), (**D**), *ste3∆-2* (SEC853), and (**E**) *dmc1∆* (SEC866) deletion mutants.

grown alone. Indeed, when the *mfα∆* and *ste3∆* mutants were cultured together, significantly more spores were observed than in either mutant (*Figure 6A and B*). This outcome could be the result of two scenarios: (i) mating (cell–cell fusion) had occurred between the two mutant strains, leading to complementation of *mfα∆* or *ste3∆* mutations by the corresponding wild-type alleles, or (ii) the pheromone secreted from the *ste3∆* mutant was being bound by the active *STE3* receptor in the *mfα∆* mutant, rescuing the defect.

To find out which of these processes occurred, spores from single basidia were dissected, and PCR was used to determine whether wild-type or mutant *MFα* and *STE3* alleles were present in the progeny. Upon the analysis of spores from 12 independent basidia, we found that both scenarios seemed to be occurring (*Figure 6C*). Most of the basidia analyzed (n = 8; basidia 2–6 and 9–11; ~67%) yielded progeny with different genotypes regarding the *MFα* and *STE3* loci, implying that mating between the two mutant strains and meiosis had occurred. Analysis of three other basidia (basidia 7, 8, and 12; 25%) showed, however, that the progeny had the same genotype as the *mfα∆* mutant parent (*Figure 6C*), indicating that exogenous pheromone from the *ste3∆* mutant may have stimulated basidia maturation and sporulation of the *mfα∆* mutant in trans. Lastly, spores recovered from basidia 1 were all *ste3∆*. This progeny may have originated from one of those rare basidia that could reach maturation and undergo sporulation, which were observed in less than 2.6% of the basidia surveyed in the *ste3∆* mutant across all experiments. Alternatively, it could be the outcome of a process analogous to pseudosexual reproduction recently characterized in *C. neoformans*, where one of the two parental nuclei is lost after mating, giving rise to uniparental progeny (*Yadav et al., 2021*).

## *C. depauperatus* undergoes intra-strain, but not inter-strain, sexual reproduction

After establishing that *C. depauperatus* is engaging in a sexual cycle involving meiosis, we sought to isolate progeny following intra- or inter-strain genetic crosses of CBS7841 and CBS7855. First, we developed an assay to readily identify recombinant progeny by isolating *C. depauperatus* strains carrying mutations easily detectable by selection on drug-containing media. To accomplish this, strains CBS7841 and CBS7855 were subjected to exposure to compounds in which mutations in target genes would lead to resistance. Sensitivity to two drugs was observed: canavanine and 5-flucytosine (5-FC). Loss-of-function mutations in the *CAN1* gene (encoding a plasma membrane arginine permease) confer resistance to canavanine, and loss-of-function mutations in any of five known genes (*FUR1, FCY1, FCY2, UXS1*, or *TCO2*) can confer resistance to 5-FC in *C. deuterogattii* and other fungi (*Srb, 1956*; *Song et al., 2012*; *Billmyre et al., 2020*). Two canavanine-resistant strains and one *FUR1* (encoding an uracil phosphoribosyltransferase) mutant strain resistant to 5-FC were isolated in the CBS7841 background (*can1-1, can1-2*, and *fur1-1*), and one canavanine-resistant strain and one *FUR1* mutant strain were isolated in the CBS7855 background (*can1-3* and *fur1-2*) (*Figure 7A and B*). We note that these mutations both confer recessive drug resistance; thus, in crosses of *can1* and *fur1* mutants, this allows haploid meiotic F1 progeny to be selected by virtue of resistance to both drugs, whereas any diploid fusion products (*can1/CAN1 FUR1/fur1*) would be sensitive to both drugs as a result of complementation.

Next, we isolated double-drug-resistant progeny by co-culturing the *can1* and *fur1* mutants and selecting for progeny resistant to both canavanine and 5-FC. Intra-strain crosses (CBS7841 *can1-2* × CBS7841 *fur1-1* or CBS7855 *can1-3* × CBS7855 *fur1-2*) and inter-strain crosses (CBS7841 *can1-2* × CBS7855 *fur1-2* or CBS7841 *fur1-1* × CBS7855 *can1-3*) were co-cultured in non-selective conditions and cells/spores were then transferred to medium containing both drugs as illustrated in *Figure 7C*. We employed PCR amplification of the *CAN1* and *FUR1* genes paired with restriction enzyme digestion (PCR-RFLP) (*Figure 7A and B*) to determine whether wild-type, *can1*, or *fur1* mutant alleles were present in the double-drug-resistant isolates. When PCR-RFLP analysis did not match the parental genotypes at both loci, the isolate was not scored as a progeny despite being double-drug resistant. Rather than cell fusion and meiosis generating the double *can1 fur1* mutant resistant to both canavanine and 5-FC, Sanger sequencing revealed that in such cases a spontaneous mutation had arisen on the background of the already known *can1* or *fur1* parental mutations (*Figure 7—figure supplement 1*). Therefore, only resistant isolates that contained the parental *can1* and *fur1* mutant alleles upon validation by PCR-RFLP were scored as a recombinant progeny. Using this screening strategy, we could find double-drug-resistant isolates containing both *can1* and *fur1* mutant alleles in isolates resulting

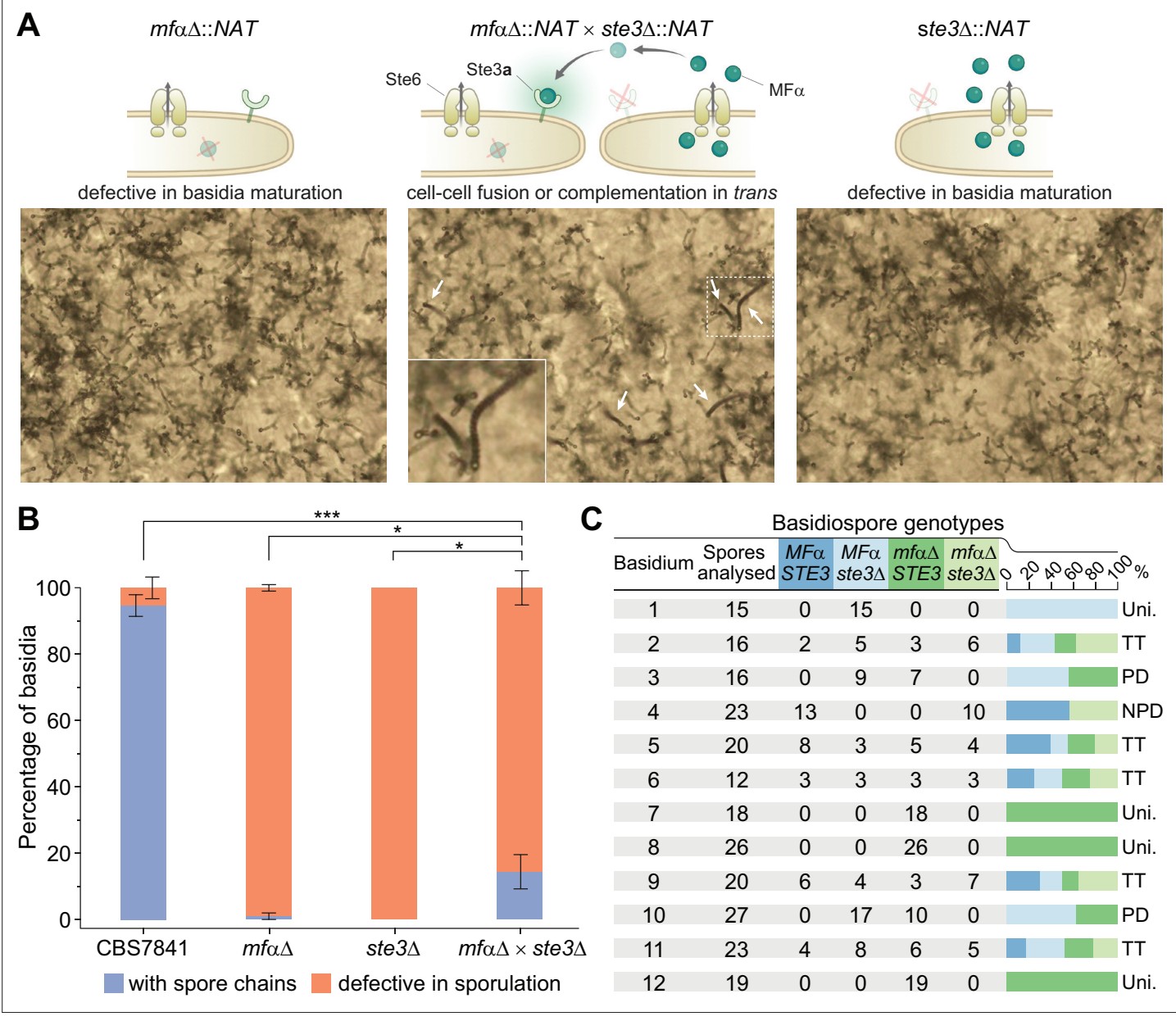

**Figure 6.** Sporulation is partially restored in *mfα∆ × ste3∆* co-cultures. (**A**) Illustration and light microscopy images of the *C. depauperatus mfα∆* mutant (left), *ste3∆* mutant (right), and a co-culture of the *mfα∆* and *ste3∆* mutants (middle). Images were taken following 2weeks of incubation on Murashige–Skoog (MS) medium. White arrows indicate spore chains, and a zoomed-in view is shown in the inset. (**B**) Quantification of basidia-producing spores and basidia defective in sporulation in wild-type (CBS7841), *mfα∆* and *ste3∆* single mutants, and the *mfα∆ × ste3∆* co-cultures. In each case, ~100 basidia across at least three independent images were quantified, and the percentage of basidia with spores (blue) and those defective in sporulation (i.e., bald basidia; orange) is shown. Error bars represent standard error of the mean. Statistical significance was determined with a one-way ANOVA and Tukey's post hoc test. *Significant at $p<0.05$; ***significant at $p<0.0001$ (see ***Figure 6—source data 1***). (**C**) Table summarizing genotypic analysis of basidiospores dissected from 12 independent basidia of the *mfα∆ × ste3∆* cross. The germination rate of the dissected spores was 100%. Presence of wild-type *MFα* and *STE3* or mutant *mfα∆* and *ste3∆* alleles was determined by in-gene and junction PCRs. Each basidium was scored as parental ditype (PD), nonparental ditype (NPD), tetratype (TT), and uniparental (Uni.) based upon the genotype of parental and recombinant spores they comprised. Phenotypic and genotypic analyses of each dissected spore are available in ***Figure 6—source data 2***. Illustrations in panel (**A**) were produced with https://biorender.com/.

The online version of this article includes the following source data and figure supplement(s) for figure 6:

**Source data 1.** Frequency of basidia defective in sporulation (bald basidia) and basidia with spores in *C. depauperatus* wild-type (CBS7841), *mfα∆* and *ste3∆* single mutants, and the *mfα∆ × ste3α∆* co-cultures mutant strains and one-way ANOVA and Tukey's HSD post hoc statistical tests for frequencies of sporulating basidia.

*Figure 6 continued on next page*

Figure 6 continued

**Source data 2.** Phenotyping and genotyping analyses of the progeny derived from *mfα*Δ × *ste3*Δ co-cultures.

**Figure supplement 1.** *C. depauperatus* mutant strains do not undergo sporulation in confrontation assays.

from intra-strain crosses of CBS7841 (*can1-2* × *fur1-1*) and CBS7855 (*can1-3* × *fur1-2*) (*Figure 7D*). In sharp contrast, none of the double-drug-resistant isolates resulting from inter-strain mating had both expected mutant parental alleles (*Figure 7D*), suggesting that some form of prezygotic (e.g., cell–cell fusion impairment) and/or postzygotic incompatibilities (e.g., the accumulation of genetic differences that could compromise meiosis) may already exist between the two strains.

## Evidence of meiotic recombination along chromosome 3 in CBS7841

In both CBS7841 and CBS7855, the *CAN1* and *FUR1* genes reside on different chromosomes: *CAN1* is on Chr 5 and *FUR1* is on Chr 2 (*Figure 8A*). Therefore, the PCR-RFLP analysis of the double-drug-resistant isolates (*Figure 7*), as well as the analysis of the *mfα*Δ × *ste3*Δ progeny (*Figure 6*), only showed that following cell fusion, independent assortment of the chromosomes had taken place. To ascertain whether the exchange was a sexual or a parasexual event, we sought to assess whether recombination in the genome had occurred. If crosses between CBS7841 and CBS7855 had produced recombinant progeny, the number of SNPs throughout the genome accounting for the 2% divergence between the strains would have been sufficient to build high-resolution meiotic maps. However, no such progeny was isolated in our crosses.

As an alternative approach, we attempted to mutagenize the CBS7841 and CBS7855 *can1* mutant strains to induce additional variants across the genome; the genetic marks could then be followed in progeny from intra-strain crosses to track recombination. However, the most mutations we could induce in a viable strain following UV mutagenesis were three nucleotide changes on the CBS7841 *can1-2* background (strain SEC747). Coincidentally, two of these mutations were spaced ~715 kb apart on the long arm of Chr 3, and the other on Chr 7 (*Figure 8A and B*).

Though only two genetic markers on a chromosome are not sufficient to finely map recombination along the chromosome, we wanted to determine whether any level of meiotic recombination could be detected between these two loci. For this purpose, the CBS7841 *can1-2* mutagenized strain (SEC747) was co-cultured with CBS7841 *fur1-1* (SEC631), and 10 independent double-drug-resistant isolates were analyzed by PCR-RFLP of the *CAN1* and *FUR1* loci, and Sanger sequencing was employed to interrogate the loci of the other mutation sites on Chr 3 and 7. In accordance with their double-drug-resistant phenotype, all progeny analyzed inherited both *can1-2* and *fur1-1* mutant loci (*Figure 8C*). For the two UV-induced mutations on the long arm of Chr 3 (designated as UV1 and UV2 in *Figure 8*), 4 out of 10 progeny had recombinant genotypes, harboring only one of the two mutations (*Figure 8C*). For instance, progeny number 3 inherited the UV2 mutation from SEC747 and the wild-type allele at the UV1 locus, suggesting that at least one crossover has occurred between these two regions. Using the data from the recombinant progeny and location of mutant alleles on Chr 3, we calculated the genetic distance between the alleles as 17.88 kb/cM (*Figure 8A*). Although this is greater than the estimates in other *Cryptococcus* species, which vary between 4.69 and 13.2 kb/cM (*Marra et al., 2004*; *Sun et al., 2014*; *Sun et al., 2017*; *Roth et al., 2018*), our analysis only utilized two markers along the entirety of the chromosome and therefore genetic distances likely have been underestimated as multiple crossover events between distant markers may skew these results.

## Discussion

Genetically controlled self-incompatibility systems are thought to have evolved to prevent inbreeding and promote outcrossing, but seeking a compatible mate is not always an easy task. This hurdle is particularly relevant for species with low population densities or that have spatially structured populations where dispersal between different patches is limited (*Murtagh et al., 2000*; *Busch and Delph, 2012*). Such a scenario of high-cost to mate finding has been proposed as an explanation for the emergence of reproductive mechanisms that overcome the low encounter rate, including hermaphroditism in many animals and plants, and homothallism in fungi, both systems allowing for reproductive assurance and the persistence of populations in a specific environment where mates are scarce or unavailable.

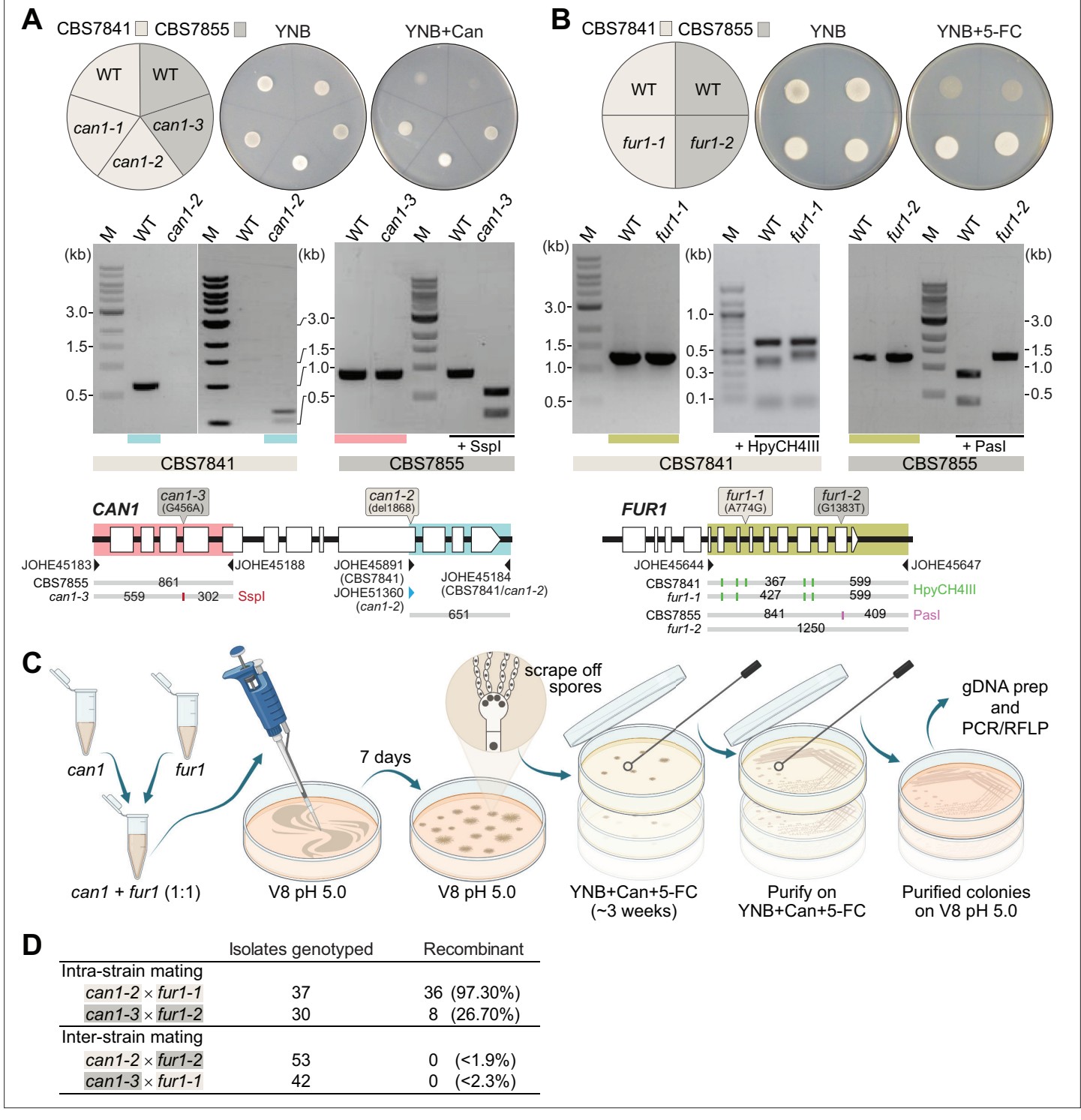

**Figure 7.** Analysis of *C. depauperatus can1* and *fur1* mutants, and mating in intra- and inter-strain crosses. Phenotypic and genotypic analysis of (**A**) *can1* and (**B**) *fur1* mutants. Top panels: wild-type and UV-induced *can1* mutants on YNB and YNB+ 60µg/mL canavanine (Can), and wild-type (WT) and spontaneous *fur1* mutants on YNB+ 100µg/mL 5-flucytosine (5-FC). Bottom panels: PCR and RFLP analyses of wild-type, and mutant *can1* and *fur1* alleles. The different *CAN1* and *FUR1* spontaneous mutations are depicted on the top of each gene (see 'Materials and methods' for details). (**C**) Schematic of mating assays with the *can1* and *fur1* mutants. (**D**) Mating assessment in CBS7841 and CBS7855 *can1* and *fur1* mutants when crossed with themselves (intra-strain crosses) and with each other (inter-strain crosses). For the inter-strain crosses, we assumed the possibility that the next isolate to be analyzed could be recombinant and thus the frequencies (shown in parentheses) were calculated as being<1/54 (*can1-2* × *fur1-2*) or<1/43 (*can1-3* × *fur1-1*).

*Figure 7 continued on next page*

*Figure 7 continued*

The online version of this article includes the following source data and figure supplement(s) for figure 7:

**Source data 1.** Source raw data for *Figure 7A and B* (raw images of gels).

**Figure supplement 1.** Sanger sequencing of *CAN1* and *FUR1* loci from double-drug-resistant isolates recovered from *C. depauperatus* intra- and inter-strain crosses.

In this study, we present four lines of evidence that *C. depauperatus* is continuously undergoing sexual reproduction. The first evidence is the significantly increased expression of key mating and meiosis genes in conditions that promote sporulation. Second, the isolation of double-drug-resistant progeny from co-cultures of *can1* and *fur1* single-drug-resistant mutants showed exchange of genetic material occurred after co-incubation of marked parental strains. Third, we identified that this exchange of genetic material most likely involves mating, cell–cell fusion, karyogamy, and meiosis as demonstrated by the fact that sporulation, and thus the production of recombinant progeny, was severely impaired upon deletion of key components required for the pheromone signaling cascade (*STE3*) or completely abolished when deleting a meiotic-specific recombinase (*DMC1*). Fourth, we detected evidence of recombination on Chr 3 in CBS7841 upon analysis of the segregation patterns of UV-induced single-nucleotide variants. Taken together, these findings provide robust evidence of meiotic sexual reproduction, and we propose that *C. depauperatus* represents the first obligately sexual homothallic species to be identified in the *Cryptococcus* genus.

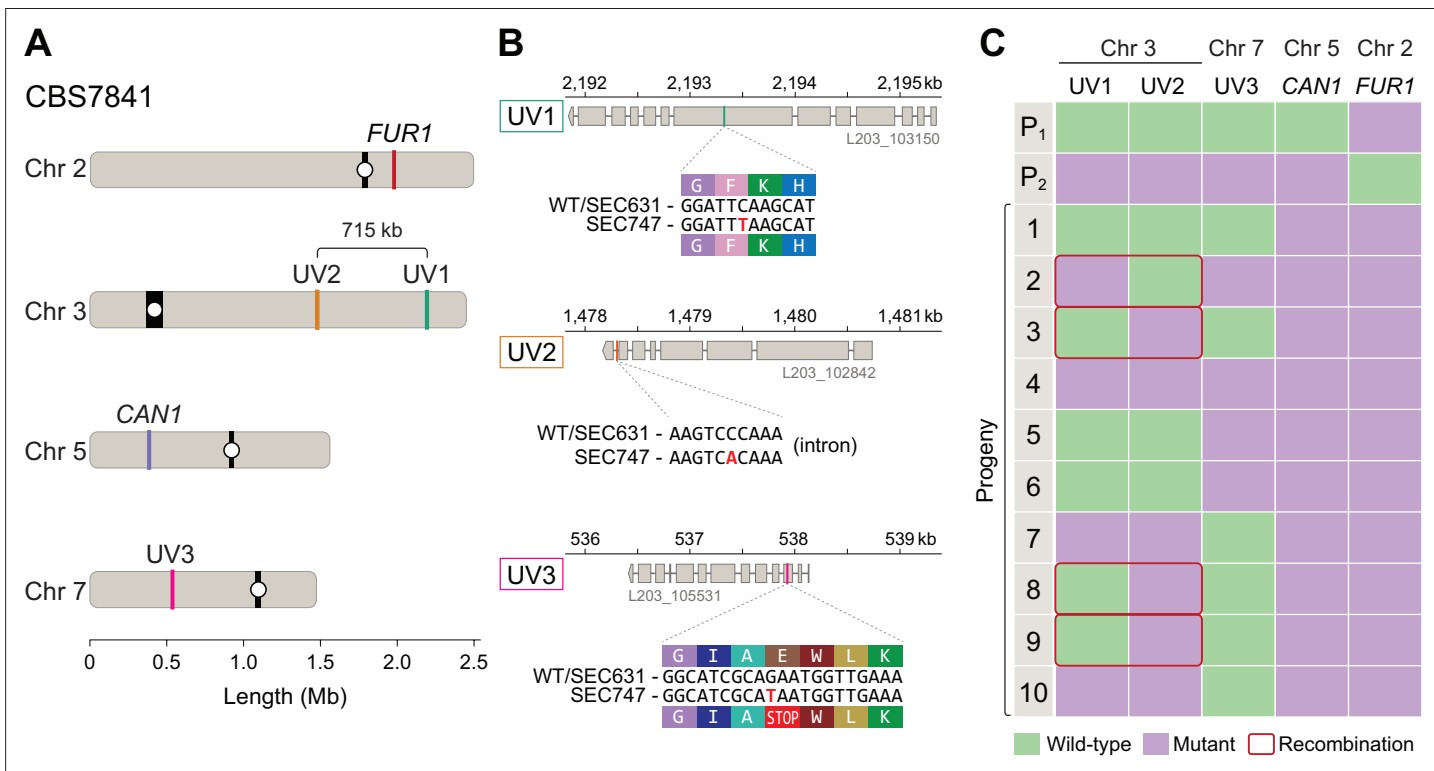

**Figure 8.** Meiotic mapping of UV-induced mutations in recombinant progeny. (**A**) Chromosome locations of the *FUR1* and *CAN1* genes and the three new mutations induced by UV irradiation (UV1, UV2, and UV3). Two of the mutations (UV1 and UV2) are spaced~715kb apart on Chr 3. (**B**) Gene models carrying the UV1, UV2, and UV3 mutations in isolate SEC747. Exons are shown as gray rectangles, while the introns are shown as gray horizontal lines. The UV1, UV2, and UV3 mutations are single-nucleotide changes, respectively, within the seventh exon of the gene L203_103150, the last intron of gene L203_102842, and third exon of gene L203_105531, the latter mutation leading to an early stop-gain. (**C**) Results of PCR and Sanger sequencing analyses used to determine inheritance of wild-type and mutant alleles in 10 progeny from a cross between isolates SEC747 and SEC631. Wild-type and mutant alleles are colored green and purple, respectively. The parent genotypes of UV1, UV2, and UV3 loci are either all wild-type alleles (P1, isolate SEC631) or all mutant alleles (P2, isolate SEC747). All progeny analyzed inherited both parental *can1-2* and *fur1-1* mutant loci, providing a double-drug-resistant phenotype. 4 out of 10 progeny had combinations of wild-type and mutant alleles on Chr 3 that differ from either parent, suggesting that meiotic recombination occurred between the two markers.

Although undergoing a meiotic cycle, our genomic analyses also revealed that two meiotic genes, *MSH4* and *MSH5*, were lost in *C. depauperatus* while being retained in other *Cryptococcus* species. In *S. cerevisiae*, Msh4 and Msh5, along with other factors, repair most (~80%) of the double-strand breaks (DSBs) initiated by Spo11 as crossovers that show interference (a.k.a. class I crossovers) (de los *de los Santos et al., 2003*; *Argueso et al., 2004*). Thus, the loss of *MSH4* and *MSH5* in *C. depauperatus*, possibly along with loss of class I crossover formation and crossover interference, indicates that the crossover homeostasis machinery might not be completely intact in this species. If confirmed, such scenario may be analogous to the loss of Msh4 and Msh5 in the fission yeast *Schizosaccharomyces pombe*, along with class I crossover formation and crossover interference losses (*Hollingsworth and Brill, 2004*; *Malik et al., 2007*), or parallel the loss of class I crossovers via loss-of-function mutations in Msh4 and/or Msh5 in model species as diverse as *S. cerevisiae*, *Caenorhabditis elegans*, and *Arabidopsis thaliana*, where such events result in fewer crossovers, and those that remain are interference-independent (i.e., produced by a pathway involving instead the Mus81-Mms4/Eme1 protein complex) (*Zalevsky et al., 1999*; *Kelly et al., 2000*; *Higgins et al., 2004*; *Hollingsworth and Brill, 2004*).

Homothallism in fungi comes in many distinct forms. The simplest configuration is a single genome that encodes compatible sets of mating-type genes, either fused or unlinked (primary homothallism). Another strategy involves mating-type switching, either bidirectional or unidirectional, which leads to a mixed population of cells of opposite mating types that can mate. A third and the most recent form of homothallism to be described is unisexual reproduction where cells of the same mating type can undergo sexual reproduction despite having genes of only one mating type (reviewed in *Ni et al., 2011*; *Roach et al., 2014*; *Fu et al., 2015*; *Wilson et al., 2015*; *Hanson and Wolfe, 2017*; *Wilson et al., 2021*). Some other fungal species, referred to as pseudo-homothallic, can complete the sexual cycle without the apparent need for cell–cell fusion, but this is achieved by packaging two opposite mating-type nuclei derived from the same meiosis into the same cell (*Lin and Heitman, 2007*).

Our study reveals that homothallism in *C. depauperatus* is independent of the homeodomain transcription factors Sxi1/HD1 and Sxi2/HD2 as these genes are completely absent in the genomes of the two strains. Rather, it seems to be orchestrated by the expression in the same genome of a single interacting mating receptor (Ste3**a**)/pheromone ligand (MFα) pair that signals through a similar mating pathway characteristic of heterothallic *Cryptococcus* species. This is similar to findings from the ascomycete fungi *Sordaria macrospora*, *N. crassa*, and *Trichoderma reesei*, where one compatible pheromone/pheromone-receptor pair in mating partners seems to be necessary and sufficient for sexual development (*Mayrhofer et al., 2006*; *Kim et al., 2012*; *Seibel et al., 2012*).

Interestingly, despite a protein sequence that differs by two amino acids, introduction of the *C. depauperatus* MFα pheromone into a *C. neoformans* MAT**a** strain induced robust hyphal formation, and this self-filamentous phenotype was abolished by deletion of either *STE6* (encoding the pheromone exporter) or the *STE3***a** gene (encoding the α-pheromone receptor). Likewise, deletion of the *MFα* and *STE3***a** genes in *C. depauperatus* resulted in severe defects in basidia maturation and sporulation, but the *C. depauperatus mfα*Δ mutant, while unable to produce the pheromone itself, could sense the pheromone secreted from the *ste3*Δ mutant in a co-culture and sporulation was partially restored. Together, this indicates that the mechanisms for pheromone export and sensing between the two species are largely conserved and reveals that pheromone/receptor recognition can transcend species boundaries.

Intriguingly, the pheromone/receptor autocrine system of *C. depauperatus* seems to control the later stage of sexual development (basidia maturation), rather than hyphal development as it does during *C. neoformans* and *C. deneoformans* heterothallic mating. Indeed, we showed that the absence of the mating pheromone or the pheromone receptor did not result in obvious impairment of self-filamentation, except for an interference in hyphal polarity resulting in tip splitting. This is similar to previously described pheromone and receptor gene deletion mutants in the filamentous ascomycetes *S. macrospora* and *Podospora anserina*, as well as deletion of both receptor genes in *Aspergillus nidulans*, which cause no changes in hyphal growth (*Seo et al., 2004*; *Coppin et al., 2005*; *Mayrhofer et al., 2006*). Conversely, basidia maturation was severely impaired in *C. depauperatus ste3*Δ, *mfα*Δ, and *ste3*Δ *mfα*Δ mutants. While the reasons underlying such observations are presently unclear, it is important to note that some of the upstream components of the mating cascade critical for the paracrine induction of hyphal formation and dikaryon maintenance during heterothallic reproduction in *C. neoformans* and *C. deneoformans* (including MFα, Ste3, and Ste6) are seemingly bypassed in

monokaryotic hyphal development and sporulation during unisexual reproduction in *C. deneoformans* (*Gyawali et al., 2017*; *Tian et al., 2018*). A reasonable explanation as to why the intercellular regulation mediated by pheromone is not strictly required for unisexual reproduction in *C. deneoformans* stems from the fact that the unisexual process in most cases occurs independently of cell–cell fusion, with cells transitioning to a diploid state either through cell fusion-independent karyogamy (minor route) or endoreplication (major route) (*Fu and Heitman, 2017*), and *C. depauperatus* may have adopted similar mechanisms.

Under this scenario, we posit that the loss of the homeodomain genes *SXI1* and *SXI2* in *C. depauperatus* may have enabled additional changes in the regulation of the mating pathway and some genes have possibly taken on additional regulatory roles downstream of the mating pheromone cascade, including in basidia maturation, meiosis, and sporulation. Two possible candidate genes that could have gained additional regulatory roles include homologs of Mat2, an HMG transcription factor, and Znf2, a zinc finger transcription factor, which are central components governing opposite and unisexual mating in *C. deneoformans* (*Lin et al., 2010*). However, we did not observe an upregulation of these genes under sporulation conditions. Hence, homothallism in *C. depauperatus* does not completely bypass the requirements for heterothallic sexual reproduction in that it requires the interplay between a compatible pheromone/receptor pair of opposite mating types. These findings show an interesting parallel with studies in *A. nidulans* that have also revealed that homothallism in this species does not bypass the requirements for partner signaling, but instead requires activation of the heterothallic signaling pathway for sexual development in a single individual (*Paoletti et al., 2007*). Future studies should address whether other aspects of the mating pathway have been rewired in *C. depauperatus* due to the loss of the homeodomain transcription factors.

Our genomic and phylogenetic analysis revealed that the predicted *MAT* locus structure of *C. depauperatus* has been extensively remodeled compared to other *Cryptococcus* species. While *C. depauperatus* has retained genes necessary for sexual reproduction, some of the key genes, including the *MFα* gene encoding the mating pheromone, are found outside of the predicted *MAT* region. In heterothallic basidiomycete fungi, one or more pheromone genes are invariably linked to a pheromone receptor of the same mating specificity (*Coelho et al., 2017*) and maintained in this way due to suppression of recombination, which in some fungal lineages has been established hundreds of millions of years ago (*Devier et al., 2009*). We hypothesize that the *MAT* gene cohort in *C. depauperatus* has been dramatically rearranged due to reduced selective pressure to maintain these genes in tight linkage, likely associated with the development of self-compatibility. Additionally, all *Cryptococcus* species so far examined encode multiple identical copies of the *MFα* or *MF***a** pheromone genes per mating type to increase the production of the mating pheromone per cell, and this is thought to enhance the functionality of the poorly diffusible small lipophilic peptides (*Akada et al., 1989*). In contrast, the presence of a single *MFα* gene copy in the *C. depauperatus* genome indicates that selection has acted to reduce the energetic burden of having extra gene copies without a direct benefit. Indeed, intercompatible MFα and Ste3**a** proteins are readily produced by the same cell, and thus, as our confrontation assays between *mfα*Δ and *ste3*Δ mutants suggested, long-distance communication is unlikely to be favored. Nevertheless, this system still allows admixture of genetic diversity via outcrossing.

Elucidating the genomic regions conferring sexual identity across a phylogeny can be a powerful approach to studying the mating-type loci structure and associated transitions between different modes of reproduction. The *MAT* locus identified on Chr 4 in both strains contains genes associated with *C. amylolentus P/R* and *HD* loci reminiscent of a past fusion event between the two regions in *C. depauperatus*. This event could have occurred in the common ancestor giving rise to *C. depauperatus* and the *Cryptococcus* pathogenic complex, or evolved instead independently in the two lineages. Our analyses cannot exclude the possibility as well that the two genomic regions in *C. depauperatus* fused together via chromosomal translocation only after homothallism was attained. Although there are several possible models to explain how the observed configuration of canonical *MAT* locus genes could have evolved from either a bipolar or tetrapolar mating system, we present two that we consider the most parsimonious given the present data and analyses (*Figure 9*).

Both models envision an ancestral state with two cells of the opposite mating-type fusing to produce a self-filamentous diploid. If the ancestor of *C. depauperatus* had a bipolar mating configuration (*Figure 9A*), then the diploid cell could have undergone a round of meiosis, resulting in an

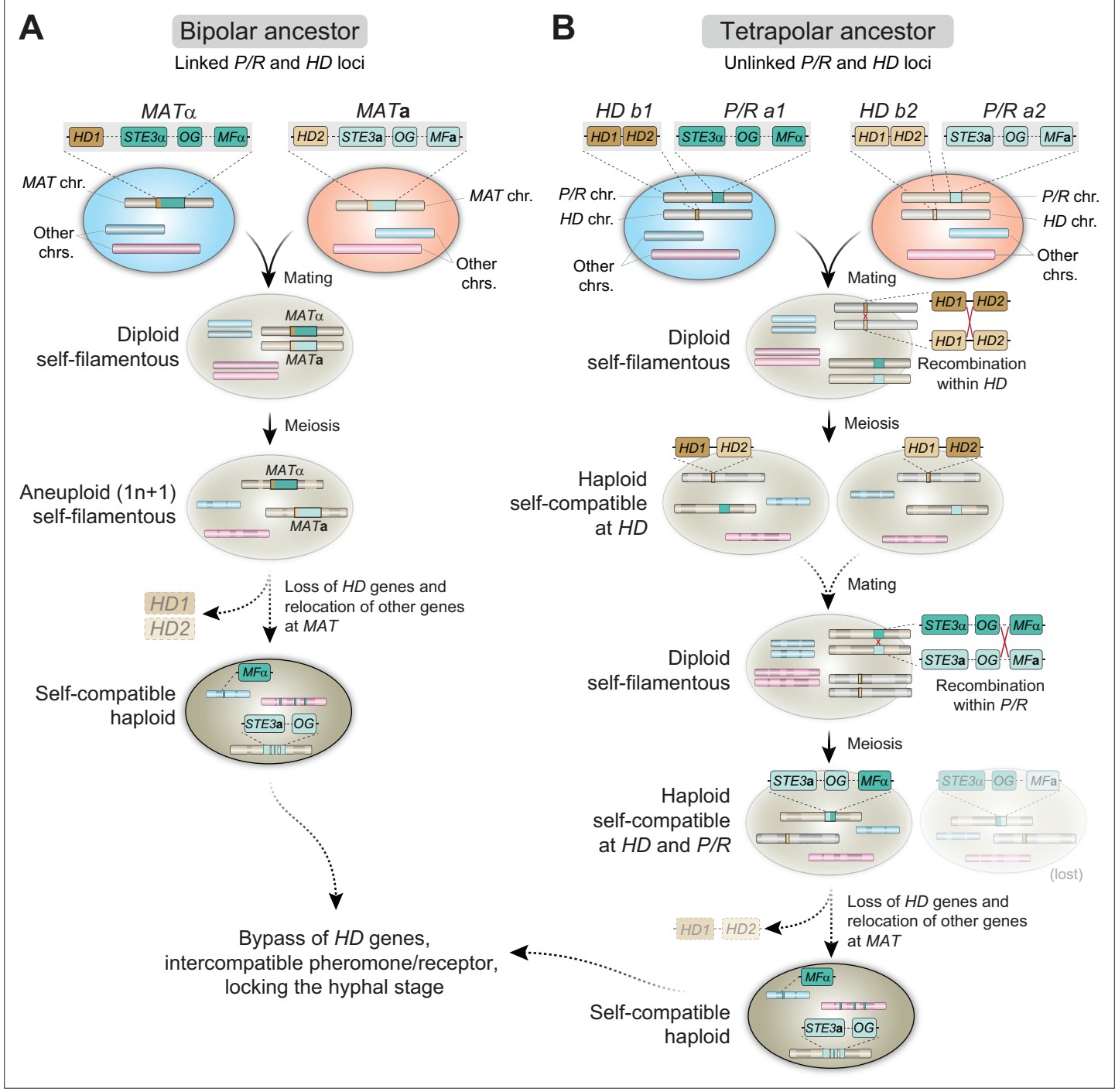

**Figure 9.** Proposed models for the evolution of *MAT* and homothallism in *C. depauperatus*. (**A**) Following cell fusion of two haploid cells with opposite mating types with linked *P/R* and *HD* loci (bipolar configuration), a self-filamentous diploid cell was formed. During meiosis, one set of homologous chromosomes, except that containing the *MAT* locus, was lost, resulting in an F1 aneuploid 1n + 1 progeny that is still self-fertile. The need for the *HD* genes is bypassed and the genes were eventually lost. As selective pressure to maintain genes in tight linkage at *MAT* was reduced due to self-compatibility, some genes from the *MAT*α loci became unlinked and dispersed throughout the genome via rearrangements and/or transposition. (**B**) Cells of opposite mating type carrying unlinked *P/R* and *HD* loci (tetrapolar configuration) fused to produce a self-filamentous diploid. During meiosis, a recombination event between *HD1* and *HD2* led to the formation of compatible *HD1/HD2* allele pairs whose gene products could heterodimerize and form an active transcriptional regulator. In a subsequent mating event involving one or both such individuals, a recombination event at the *P/R* locus caused reciprocal exchange between select *P/R a1* (including *MF*α) and *P/R a2* genes and gave rise to two haploid, self-filamentous progeny both now capable of responding to their own pheromone. In such a scenario, the cell harboring the *STE3*a and *MF*α genes may have subsequently lost the compatible *HD1/HD2* gene pair, and some *MAT*α genes became unlinked and dispersed throughout the genome. The cell harboring the *STE3*α and *MF*a genes was either lost or outcompeted. *OG* stands for other genes typically included at the *P/R* locus beyond mating pheromones and receptors.

F1 aneuploid 1n + 1 progeny containing the genome of the *MAT***a** parent plus the *MAT* chromosome of the *MAT*α parent. This progeny would be self-filamentous and potentially able to undergo a sexual cycle. If the ancestor had instead a tetrapolar mating configuration (*Figure 9B*), two intra-*MAT* recombination steps are required to achieve homothallism. First, a recombination event at the *HD* locus would bring together compatible *HD1* and *HD2* alleles, whose gene products could heterodimerize and form an active transcription regulator. The haploid individuals carrying this compatible *HD* pair would shift from compatibility determined by two sequential checkpoints (*P/R* and *HD*) to compatibility determined by a single mechanism (*P/R*), representing the first step in a transition from heterothallism to homothallism. A complete transition to homothallism could then be achieved in a subsequent mating event by uniting in the same genome-compatible pheromones and receptors via recombination; the isolates resulting from such event would respond to their own pheromone. In both models, we posit that the selective pressure to maintain an intact *MAT* locus is likely relaxed once self-compatibility is established, allowing chromosomal rearrangements to distribute some of the *MAT* genes throughout the genome without a concomitant fitness disadvantage. The *SXI1*/*HD1* and *SXI2*/*HD2* genes are eventually lost once their function is bypassed or supplanted by other genes.

The varied mechanisms via which homothallism has evolved and is maintained in fungi suggest that it represents an evolutionarily significant mating strategy. Examples from other fungi provide additional insights as to how the homothallic lifestyle of *C. depauperatus* evolved. While the number of representative samples is presently limiting, increased sampling efforts over a broader range of niches, including surveys for mycoparasites, may prove a good strategy to uncover additional isolates of the same or from closely related species, as a recent study suggests (*Guterres et al., 2021*). A careful characterization of the ability to sexually reproduce will ultimately shed light on how this unique continuously active sexual development program emerged, making it a contrasting *Cryptococcus* species. Hence, *C. depauperatus* provides another vantage point and an opportunity to further understand both the evolution of sexual reproduction in the pathogenic and saprobic *Cryptococcus* lineages, and the frequent transitions between modes of sexual reproduction that have occurred throughout the fungal kingdom.

## Materials and methods
### Strains, growth conditions, and primers
*C. depauperatus* strains were sub-cultured from –80°C onto V8 pH 5.0 (5% V8 juice, 0.05% w/v KH$_2$PO$_4$, 4% agar) or Murashige–Skoog (MS) medium plates, and maintained on V8 pH 5.0 medium at room temperature thereafter unless otherwise noted. *C. neoformans* strains were maintained on yeast, peptone, dextrose (YPD) and incubated at 30°C after subculturing from freezer stocks, and then stored at 4°C. Both *C. depauperatus* and *C. neoformans* strains are stored long term in 25% glycerol stocks at –80°C. All strains and primers used in this study are listed in *Supplementary file 1* and *Supplementary file 2*, respectively.

### DNA extraction, CHEF analysis, and whole-genome sequencing of *C. depauperatus* strains
To isolate genomic DNA, cells from CBS7841 and CBS7855 were scraped from V8 pH 5.0 plates and collected in a 50mL conical tube. Cells were frozen at –80°C for at least 1 hr and lyophilized overnight. Genomic DNA was extracted from the dried cell debris using a modified cetyltrimethylammonium bromide (CTAB) (*Pitkin et al., 1996*). To recover high-molecular-weight DNA samples, precipitated DNA was spooled out using a glass rod, and the size and integrity of the DNA were confirmed by clamped homogeneous electrical field (CHEF) electrophoresis. Whole-genome sequencing was performed with Illumina and Oxford Nanopore Technologies (ONT). For Illumina sequencing, two libraries were constructed. A short-fragment library was prepared from 100ng of genomic DNA sheared to~250bp using a Covaris LE instrument and adapted for sequencing as previously described (*Fisher et al., 2011*). A 2.5kb jumping library was prepared using the 2-to-5-kb insert Illumina Mate-pair library prep kit (V2; Illumina). These libraries were sequenced at the Broad Institute Genomics Platform on an Illumina HiSeq 2000 system to generate paired 101-base reads. For nanopore sequencing, libraries of the two strains were barcoded and constructed as per the manufacturer's instructions using the SQK-LSK109 and EXP-NBD103/EXP-NBD104 kits. DNA samples were pooled together and run

for 48 hr in a single R9 flow cell (FLO-MN106) in a MinION system controlled by MinKNOW. Base-calling was performed with Guppy v3.3.3 GPU mode and the parameters "-c dna_r9.4.1_450bps_hac.cfg --min_qscore=7 --qscore_filtering". The resulting "PASS" reads were de-multiplexed with guppy_barcoder with parameters "--barcode_kits EXP-NBD104 --records_per_fastq 0 --trim_barcodes". CHEF electrophoresis analyses for separating individual chromosomes were carried out as previously described (**Sun et al., 2017**) using switching times of 120–260s for smaller chromosomes and 560–700s for larger chromosomes.

## Genome assembly

The genomes of CBS7841 and CBS7855 were assembled from nanopore long-reads with Canu v1.8 (**Koren et al., 2017**) using default parameters and an estimated genome size of 16.5 Mb. Sequence accuracy of the draft assemblies was improved by correcting errors using Nanopolish v0.11.2 (https://github.com/jts/nanopolish, **Simpson et al., 2019**) and with five rounds of Pilon v1.22 polishing with the "--fix all" parameter (**Walker et al., 2014**) employing the Illumina reads mapped to the respective assemblies using BWA-MEM v0.7.17-r1188 (**Li and Durbin, 2009**). For each assembly, the corresponding Canu-corrected reads were aligned to the entire genome via minimap2 v2.9-r720 (**Li, 2018**) and read coverage profiles were visualized with the Integrative Genomics Viewer (IGV) v2.8.0 (**Robinson et al., 2011**) to identify large collapsed regions or mis-assemblies. Reads aligned to the CBS7841 genome revealed a region on the original contig3, corresponding to the predicted centromere, at roughly double the assembly mean depth of coverage. Viewing this region of double coverage in IGV revealed a specific contig position where nearly all reads were split-aligned, indicating missing sequence in the consensus. One ONT read that was not split-aligned showed an insertion "I" in IGV with length of 12,682 bp. Inspecting the alignments of additional ONT split-aligned reads confirmed the length and content of the missing consensus sequence. This 12,682 bp region was extracted from one of the corrected ONT reads and inserted into the original contig3 at the exact site of the split-aligned read consensus base. For CBS7855, detection of low read coverage approximately in the middle of the original contig1 was also suggestive of misassembly. In this case, the assembler had joined the end of a chromosome with the arm of another chromosome (i.e., generating a junction between a telomere and part of a centromere), most likely due to the repetitive nature of some of the centromeres and telomeres. To resolve this, two breaks were first generated on the original contig1 to remove an ~40 kb region that was duplicated at the end of the original contig8. Second, LRscaf v1.1.9 with parameters "-misl 5 -micl 1000" and SSPACE-long v1.1 with parameters "-l 5 -o 1000" were employed to scaffold the resulting contigs. Both programs produced identical results, joining the initial contig8 with one of the broken fragments of the initial contig1. Third, multiple rounds of sequence alignment were employed to repair the end of the remaining fragment of the initial contig1. After these steps, remapping of the Canu-corrected reads recovered a more uniform distribution across the genome, showing that the assembly is free from large structural errors. The resulting assemblies of CBS7841 and CBS7855 were subjected to a final polishing step with Nanopolish, and two rounds of Pilon as described above.

## Gene prediction and annotation

To minimize the prediction of false-positive gene structures in repetitive and low-complexity regions, the genome assemblies were initially soft masked for repeats by RepeatMasker version open-4.0.7 using a custom repeat library consisting of known *C. neoformans* transposable elements (**Goodwin and Poulter, 2001**) and *C. depauperatus* elements identified de novo by RepeatModeler2 (https://github.com/Dfam-consortium/TETools; **Flynn et al., 2020**). Gene models were predicted with the BRAKER2 pipeline v2.1.5 (https://github.com/Gaius-Augustus/BRAKER; **Bruna et al., 2021**) run in ETP-mode with parameters "--etpmode --softmasking --fungus". In this running mode, BRAKER employs predicted protein sets of other species processed by ProtHint and RNA-seq data for training Gene-Mark-ETP (**Buchfink et al., 2015**; **Bruna et al., 2020**). We used as input the available protein sets of *C. neoformans* H99 (**Janbon et al., 2014**) and *C. amylolentus* CBS6039 (**Sun et al., 2017**) and RNA-seq data obtained from *C. depauperatus* grown in two different conditions (see below). RNA-seq data was preprocessed using Trim Galore v0.6.5 (https://github.com/FelixKrueger/TrimGalore, **Krueger, 2019**) with default parameters except that reads shorter than 75 nt after quality or adapter trimming were discarded (parameters: "--paired --quality 20 --phred33 --length 75"). Splice read

alignment was performed with STAR aligner v2.7.4a (*Dobin et al., 2013*) (specific parameters for indexing: "`--genomeSAindexNbases 11`"; specific parameters for aligning: "`--alignIntronMin 10 --alignIntronMax 2000 --outSAMtype BAM SortedByCoordinate`") and the resulting BAM files were input to the BRAKER pipeline. In this running mode of BRAKER, intron information present in both data sources was weighted as reliable evidence and enforced for gene prediction in both GeneMark-ETP and in AUGUSTUS. Gene predictions were further refined by removing spurious gene calls identified within the predicted centromeric regions and manually adding the short mating pheromone precursor gene. tRNAs were predicted using tRNAscan (*Chan and Lowe, 2019*) and rRNAs predicted using RNAmmer (*Lagesen et al., 2007*). Genes containing PFAM domains found in repetitive elements or overlapping tRNA/rRNA features were removed. Genes were named and numbered sequentially with "locus_tag" ids for CBS7841 and CBS7855 starting at L203_100001 and L204_100001, respectively, to distinguish from the gene calls previously deposited in GenBank for the two strains under the same "locus_tag" prefix. For the protein coding gene name assignment, we combined names from HMMER PFAM/TIGRFAM, Swissprot, and KEGG products. The completeness of the genome assemblies and gene predictions was evaluated with Benchmarking Universal Single-Copy Orthologs (BUSCO, v4.0.6) based on the presence/absence of a set of predefined single-copy orthologs of other Tremellomycetes (tremellomycetes_odb10 dataset) (*Kriventseva et al., 2019*; *Manni et al., 2021*).

## Ortholog identification and phylogenomic data matrix construction

To construct the phylogenomic data matrix, single-copy orthologs were identified across selected strains/species representing all major *Cryptococcus* lineages (*C. neoformans* H99, *C. deneoformans* JEC21, *C. gattii* WM276, *C. amylolentus* CBS6039 and CBS6273, *C. floricola* DSM27421, and *C. wingfieldii* CBS7118) and the outgroup *K. mangrovensis* (strains CBS8507 and CBS10435) using OrthoFinder v2.4 (*Emms and Kelly, 2019*), with parameters "-M msa -S diamond -I 1.5 -T iqtree". The amino acid sequences of 4074 single-copy orthologs shared among all species were individually aligned with MAFFT v7.390 (*Katoh and Standley, 2013*) with arguments `--localpair --maxiterate 1000` and trimmed with TrimAl v1.4.rev22 (*Capella-Gutiérrez et al., 2009*) using the "-gappyout -keepheader" options.

## Phylogenetic analyses and evaluation of topological support

Two different analytical methods were employed to infer the evolutionary relationships among the selected taxa: (i) concatenation with gene-based partitioning and (ii) gene-based coalescence. ML phylogenetic trees were built using IQ-TREE v2.1.2 (*Minh et al., 2020b*). For the first method, a directory containing the individual protein alignments was passed to IQ-TREE with the argument "-p", which automatically loads and concatenates all alignment files into a supermatrix for partition analysis and applies an edge-linked proportional partition model to accommodates different evolutionary rates between partitions. Best-fitting amino acid substitution models were determined for each partition with the Bayesian information criterion (BIC). The best-scoring ML concatenation-based ML tree was inferred using the parameters "`--seed 12345 -B 1000 -alrt 1000 -T 12`". Support of each internal branch was assessed using 1000 replicates of the Shimodaira–Hasegawa approximate likelihood ratio test (SH-aLRT) and ultrafast bootstrap (UFboot).

For the second method, individual ML gene trees were inferred for the set of 4074 single-copy orthologs with IQ-TREE but using instead the argument "-S", which instructs the program to perform model selection and tree inference separately for each alignment. A coalescent-based species phylogeny was then inferred with ASTRAL-MP v5.15.2 (*Yin et al., 2019*) using a file containing the set of 4074 individual ML gene trees and the option "-t 2" to obtain on each branch a full set of different measurements, including quartet support for the main (q1) and alternative topologies (q2 and q3). Contracting branches with low support in gene trees has been shown to improve accuracy of coalescent-based tree inference with ASTRAL (*Zhang et al., 2017*). Therefore, branches with bootstraps < 90% were contracted with Newick utilities v1.6 (*Junier and Zdobnov, 2010*) using the options " 'i & b ≤ 90' o", resulting in poorly supported bipartitions within each gene tree becoming collapsed into polytomies. The resulting file was then run through ASTRAL as above, and local posterior probability (LPP) support values were plotted in the final tree. The best tree topologies obtained by both methods were congruent.

To quantify genealogical concordance and topological support, we analyzed each bipartition in the reference ML phylogeny by measuring the gene concordance factor (gCF) and the site concordance factor (sCF) as implemented in IQ-TREE v2.1.2 (*Minh et al., 2020a*). For every branch of the species ML reference tree, gCF describes the fraction of single-gene trees that can recover a particular branch, and sCF is defined as the percentage of decisive sites of the alignment that support a given branch in the species tree. To evaluate this, the file containing all 4074 single-gene trees was input to IQ-TREE using the specific options "`--gcf --scf` 1000 -seed 132465".

For the phylogenetic analysis of selected genes found within the *MAT* locus of *C. neoformans* (*Figure 2—figure supplement 1*), amino acid sequences were retrieved for each species, aligned, and trimmed as above, and subsequently used to infer ML phylogenies in IQ-TREE. All phylogenetic trees were visualized with iTOL v5.6.3 (*Letunic and Bork, 2021*).

## Whole-genome pairwise identity, analysis of genomic features, and synteny comparisons

The average pairwise identities between the two genomes were calculated using dnadiff (https://github.com/marbl/MUMmer3/blob/master/docs/dnadiff.README), which is a wrapper around NUCmer, included in the MUMmer package (*Kurtz et al., 2004*). Alignments were filtered with a delta filter using parameters '–1' to select 1-to-1 alignments, allowing for rearrangements and '-l 100' to select a minimal alignment length of 100 bases.

The genomic features depicted in the circos plots were determined as follows: (i) repetitive DNA content, including transposable elements, was analyzed with RepeatMasker version open-4.0.7 (using RepBase-20170127 and Dfam_Consensus-20170127) and identified de novo by RepeatModeler2 (https://github.com/Dfam-consortium/TETools, *Rosen, 2022*); (ii) centromeres were predicted on the basis of the detection of centromere-associated LTR elements previously reported in *C. amylolentus* (Tcen1 to Tcen6) (*Sun et al., 2017*) and *C. neoformans* (Tcn1 to Tcn6) (*Janbon et al., 2014*). Most of these elements mapped to the largest ORF-free region in each contig, which also coincided with regions with low/minimal transcription; (iii) the GC content was calculated in nonoverlapping 5 kb windows using a modified Perl script (gcSkew.pl; https://github.com/Geo-omics/scripts/blob/master/scripts/gcSkew, *Michigan Geomicrobiology Lab, 2019*) and plotted as the deviation from the genome average for each contig; (iv) rRNA genes (18S, 5.8S, 28S, and 5S) and tRNA genes were inferred and annotated using RNAmmer v1.2 and tRNAscan-SE v2.0, respectively; and (5) telomeric repeats were identified at the end of each contig and inspected genome-wide with EMBOSS fuzznuc (*Rice et al., 2000*) using a search pattern of 2 × TAAC(4,5), allowing for minor variation between the repeats.

Genomic synteny analyses between CBS7841 and CBS7855 were conducted with BLASTN, and the results were plotted using Circos v0.69–6 (*Krzywinski et al., 2009*). The Python application Easyfig v2.2.3 (*Sullivan et al., 2011*) was employed to produce linear synteny plots and the resulting figures were further edited for representation purposes in Adobe Illustrator.

## Read mapping and variant calling

To detect variants in *C. depauperatus* strains that underwent UV irradiation, their genomes were subjected to Illumina paired-end sequencing on a HiSeq 4000 system. Paired-end reads of 151 bp were mapped to the respective reference genome with BWA-MEM v0.7.17-r1188 with default settings. Picard tools (http://broadinstitute.github.io/picard/) integrated in the Genome Analysis Toolkit (GATK) v4.0.1.2 (*DePristo et al., 2011*) was used to sort the resulting files by coordinate, to fix read groups (modules: SORT_ORDER=coordinate; 'AddOrReplaceReadGroups') and to mark duplicates. Variant sites were identified with HaplotypeCaller from GATK using the haploid mode setting, and only high-confidence variants that passed a filtration step were retained (the "VariantFiltration" module used the following parameters: DP <20 ‖ QD <15.0 ‖ FS >60.0 ‖ MQ <55.0 ‖ SOR >4.0). The resulting files in the variant call format (VCF) were visually inspected with IGV v2.8.0. New variants detected in the mutant strains were used as markers to score recombination or independent assortment of chromosomes after meiosis.

## Divergence plots

To investigate the occurrence of highly similar genomic regions (>5 kb) between the two *C. depauperatus* strains, Illumina reads generated for CBS7855 were aligned to the CBS7841 reference

genome assembly with the methods described above. The resulting consensus genotype VCF file was converted to the FASTQ by limiting maximum depth to 200 to avoid overrepresented regions using "vcfutils vcf2fq -D 200" included in the BCFtools v1.7 package (https://github.com/samtools/bcftools/tree/develop/misc; *Danecek, 2018*). A FASTA file was then generated in which bases with quality values lower than 20 (equivalent to 99% accuracy) were soft-masked to lowercase ("seqtk seq -q 20") (https://github.com/lh3/seqtk; *Li, 2022*) and ambiguous bases were subsequently converted to an "N". Levels of divergence per site (*k*, with Jukes–Cantor correction) between the two strains were estimated with VariScan v.2.0.3 (*Vilella et al., 2005*) using a nonoverlapping sliding window of 5000 sites. The resulting data was plotted with pyGenomeTracks (*Lopez-Delisle et al., 2021*). Because highly divergent regions (particularly the centromeres) are challenging to align to a reference genome using short-read data, all divergence estimates should be regarded as minimum estimates.

## RNA extraction and RNA-seq analysis

RNA was extracted from *C. depauperatus* strains CBS7841 and CBS7855 grown in liquid culture and from solid medium. For liquid cultures, spores were scraped from V8 pH 5.0 plates and transferred to YPD liquid medium in 250 mL flasks. Liquid cultures were incubated at room temperature for 5 days with agitation. Following incubation, cultures were filter-sterilized, and the mycelia collected on the filter was scrapped off and transferred to 50 mL conical tubes. For cells harvested from solid medium, spores and mycelia were scraped from 10 V8 pH 5.0 plates, in triplicate, that had been incubated face-up in the dark at room temperature for 5 days. A cellophane membrane was placed on the agar prior to inoculation to prevent mycelia from embedding in the agar and allowing them to be harvested. Spores and mycelia from the plates were transferred to 50 mL conical tubes. Approximately 4 mL of cell mass was collected for each strain in each condition, and the rest was discarded. The 50 mL conical tubes were placed at –80°C for at least 1 hr before being lyophilized. After lyophilization, the cells were pulverized to a fine powder with 3 mm glass beads and vortexing. 4 mL of Trizol (Thermo Fisher) was added to the cell powder along with 0.8 mL of chloroform, and the mixture was centrifuged. The aqueous phase was transferred to a new tube and mixed with 2 mL of isopropanol. Each sample was then divided among six QIAGEN RNeasy mini spin columns and purified according to kit instructions. Each tube was eluted with 50 μL of DNase-free water and, for every sample, combined into a single 1.5 mL tube. RNA was treated with Turbo DNase (Thermo Fisher) according to kit instructions.

RNA sequencing was performed by Genewiz, Inc using a strand-specific protocol that included poly-A selection. Sequencing was done on an Illumina HiSeq instrument using paired-end, 150 bp reads. Trimming of sequence reads was done as described before (*Teichert et al., 2012*), and reads were mapped to the genomic sequences using HISAT2 v2.1.1 (*Kim et al., 2019*). Transcript levels were compared using the algorithms implemented in DESeq2 v1.4.5 (which report p-values that indicate statistical significance; i.e., likelihoods that a gene is not differentially expressed) (*Love et al., 2014*) and in LOX (Levels Of eXpression) v1.8 (which reports Bayesian p-values for differential expression; i.e., likelihoods that a gene is differentially expressed) (*Zhang et al., 2010*).

## Expression of *C. depauperatus* pheromone in *C. neoformans*

The *C. depauperatus* pheromone precursor encoding gene was amplified from CBS7841 using primers JOHE43884 and JOHE43885. Along with the actin (*ACT1*) promoter and terminator from *C. neoformans* H99 (amplified with primer pairs JOHE43888/JOHE43883 and JOHE43886/JOHE43889, respectively), the pheromone gene was cloned into pSDMA57, a *C. neoformans* complementation vector targeted for insertion into the safe haven locus on Chr 1 (*Arras et al., 2015*). BaeI-digested plasmid containing the cloned *C. depauperatus MFα* gene was introduced into *C. neoformans* strains YPH570 (*MAT**a** crg1Δ*) and F99 (*MATα crg1Δ*) via biolistic transformation (*Toffaletti et al., 1993*). Transformants were selected on YPD agar medium supplemented with 200 μg/mL G418 (NEO) and confirmed by PCR using primers JOHE44478 to JOHE44481. Two independent transformants in each background were selected for further analysis.

We made *C. neoformans ste3Δ* and *ste6Δ* gene deletion mutants using the efficient homologous recombination system of *S. cerevisiae* (*Ianiri et al., 2016*). The upstream region of *STE3* was amplified with primers JOHE45684 and JOHE45686, the *HYG* resistance gene was amplified from pJAF15 plasmid with JOHE45687 and JOHE45688, and the *STE3* downstream region was amplified with JOHE45689 and JOHE45690. All PCR products were transformed into *S. cerevisiae*, along with XhoI

and EcoRI digested pRS426 plasmid. Transformants were selected on synthetic dextrose without uracil (SD-ura). The full *STE3* deletion construct was amplified from *S. cerevisiae* with primers JOHE45692 and JOHE45693 and introduced into *C. neoformans* via biolistic transformation. Transformants were selected on YPD agar medium supplemented with 200 µg/mL hygromycin (HYG), and replacement of the *STE3* gene was confirmed with 5′ junction, 3′ junction, in-gene, and spanning PCRs. Similarly, to generate *ste6Δ* mutants, PCR products with primer pairs JOHE45700/JOHE45702 (*STE6* upstream region), JOHE45703/JOHE45704 (*HYG* marker), and JOHE45705/JOHE45706 (*STE6* downstream region) were transformed into *S. cerevisiae* along with XhoI and EcoRI digested pRS426 plasmid. The *ste6Δ* deletion construct was amplified from *S. cerevisiae* with primers JOHE45708 and JOHE45709 and introduced into *C. neoformans* via biolistic transformation. The s*te6Δ* deletion mutants were confirmed with 5′ junction, 3′ junction, in-gene, and spanning PCRs. All primers used in this study are listed in *Supplementary file 2*. To assess the ability of the transformants to filament, strains were inoculated onto filament agar for 2 weeks and incubated at room temperature in the dark. Images were taken with a Zeiss Axiocam 105 color camera attached to a Zeiss Scope.A1 microscope.

### *Agrobacterium tumefaciens*-mediated transformation of *C. depauperatus*

Plasmids engineered for ectopic integration and gene deletion were generated using the efficient homologous recombination system of *S. cerevisiae* and the binary vector pGI3, which can be replicated in *Escherichia coli*, *A. tumefaciens,* and *S. cerevisiae* (*Ianiri et al., 2016*). The plasmid harboring the *C. depauperatus*-specific *NAT* resistance gene was constructed as follows. First, the promoter of the actin (*ACT1*) gene (L203_102772) was amplified from *C. depauperatus* CBS7841 with primers JOHE46155 and JOHE46156 (591 bp), the terminator of the phosphoribosyl anthranilate isomerase (*TRP1*) gene (L203_102914) was amplified using primers JOHE46165 and JOHE46168 (239 bp), and the *NAT* resistance gene was amplified from plasmid pAI3 using primers JOHE46159 and JOHE46162. Secondly, these PCR products, along with BamHI and KpnI digested pGI3, were transformed into *S. cerevisiae* and combined through homologous recombination, and transformants were selected for on SD-ura (*Figure 5—figure supplement 1A*). Transformants were screened by colony PCR using primer pairs JOHE43279/JOHE50308 (5′ junction) and JOHE50309/JOHE43280 (3′ junction). Genomic DNA was isolated from transformants correct for the 5′ and 3′ junction PCRs, and PCRs of the P$_{ACT1}$-*NAT* and *NAT*-T$_{TRP1}$ junctions were checked with primers JOHE50285/JOHE41081 and JOHE40162/JOHE50286, respectively. Genomic DNA from transformants correct at all junctions was electroporated into *A. tumefaciens* strain EHA105 and transformants were selected in YT + 50 µg/mL kanamycin, and further validated using colony PCR for the 5′ and 3′ junctions. For targeted gene replacement, regions flanking the target genes (*MFα*, *STE3,* and *DMC1*) were amplified by PCR using primers that include additional sequences specific to the pGI3 plasmid and *NAT* cassette for homologous recombination. The forward primer to amplify upstream gene regions also contained homology to the pGI3, whereas the reverse primer contained homology to *NAT*. Likewise, the forward primer to amplify downstream gene regions contain homology to *NAT* and the reverse primer contained homology to pGI3 (primers listed in *Supplementary file 2*).

The transconjugation experiments with *C. depauperatus* were performed by co-incubating *A. tumefaciens* strain EHA105 carrying the desired plasmid, prepared as previously described (*Bundock et al., 1995*), and adjusted to an OD$_{600}$ of 0.6 with CBS7841 *C. depauperatus* spores freshly isolated from V8 pH 5.0 or MS plates or CBS7855 spores germinated for 24 hr in YPD liquid medium, also adjusted to an OD$_{600}$ of 0.6 (*Figure 5—figure supplement 1B*). *A. tumefaciens* and *C. depauperatus* cells were aliquoted onto a nylon membrane placed on top of induction media (MM salts, 40 mM 2-(*N*-morpholino)ethanesulfonic acid (MES), pH 5.3, 10 mM glucose, 0.5% w/v/ glycerol) containing 100 µM acetosyringone in four separate patches using varying ratios of fungal and bacterial cells: 100 µL of *A. tumefaciens* + 100 µL *C. depauperatus*, 100 µL of *A. tumefaciens* + 10 µL *C. depauperatus*, 10 µL of *A. tumefaciens* + 100 µL *C. depauperatus*, and 10 µL of *A. tumefaciens* + 10 µL *C. depauperatus*. Co-cultures of *A. tumefaciens*/*C. depauperatus* were incubated on induction medium agar for 3–10 days, after which all cells (bacterial and fungal) were scraped from the induction media and transferred to YPD medium supplemented with 200 µg/mL of cefotaxime to inhibit growth of *A. tumefaciens*, and 300 µg/mL of NAT to select *C. depauperatus* transformants. Plates were placed at 30°C until colonies appeared. Transformants with ectopic integration of the *NAT* cassette were screened using primers

JOHE40162 and JOHE41081. Gene deletion transformants were screened at the 5' and 3' junctions, as well as in-gene and spanning PCRs with primers listed in **Supplementary file 2**.

## Isolation of *C. depauperatus* mutants resistant to canavanine and 5-FC

To isolate canavanine-resistant mutants, spores from CBS7841 and CBS7855 were scraped from V8 pH 5.0 plates by pipetting 1 mL of dH$_2$O onto the cells and using a cell scraper to gently scrape spores off the plate. The spores were transferred to an YPD plate and irradiated with 10,000, 20,000, and 30,000 µJ/cm$^2$ UV with a Stratagene Stratalinker 2400. Spores were allowed to recover for 3 days before being scrapped from the YPD plate and transferred to YPD plates containing 60 µg/mL canavanine. The YPD + canavanine plates were incubated at room temperature in the dark until colonies appeared. Spores from the YPD + canavanine plates were scraped from single colonies using a toothpick and struck onto YPD + canavanine again to purify. Purified single colonies were transferred to V8 pH 5.0 plates to obtain more biomass for gDNA extraction. To isolate 5-FC-resistant mutants, spores from CBS7841 and CBS7855 were scraped from V8 pH 5.0 plates and transferred to YNB plates supplemented with 100 µg/mL of 5-FC and UV mutagenesis was not required in this case. Plates were incubated at room temperature until resistant single colonies appeared. Resistant colonies were purified on YNB + 5-FC, and purified single colonies transferred to V8 for gDNA isolation. gDNA was prepared by resuspending spores in 200 µL TENTS buffer (20 mM Tris pH 7.0, 2 mM EDTA, 500 mM NaCl, 1% Triton X-100, 1% SDS) and adding 200 µL of phenol:chloroform:isoamyl alcohol 25:24:1. A volume of approximately 200 µL of glass beads was added and tubes were bead beated for 5 min. Following bead beating, 200 µL of Tris pH 7.0 was added, and the mixture was briefly vortexed. Tubes were spun in a microcentrifuge for 5 min at 15,000 rpm. The top aqueous phase was transferred to a clean 1.5 mL tube, and DNA was precipitated with 500 µL ethanol. DNA was pelleted in a microcentrifuge for 10 min, washed with 70% ethanol, dried, and resuspended in 50 µL dH$_2$O.

To determine the mutation, the *CAN1* gene was amplified from canavanine-resistant isolates using primers JOHE45183 and JOHE45184. PCR products were run on an 0.8% agarose gel, purified, and sent for Sanger sequencing (Genewiz, Inc). Likewise, the *FUR1* gene was amplified from 5-FC-resistant mutants using primers JOHE45642 and JOHE45653, purified, and sent for Sanger sequencing. Sequencher v5.4.6 was used to analyze gene sequences and identify mutations.

## Co-culturing of *C. depauperatus* and genotypic analysis

The isolation of *can1 fur1* recombinant strains was performed as illustrated in **Figure 7C**. Spores of the *can1* and *fur1* mutant strains were scraped from V8 pH 5.0 plates using 1 mL dH$_2$O and transferred to a sterile 1.5 mL tube. The final volume in the tube was brought up to 1 mL with dH$_2$O if needed. In a clean 1.5 mL tube, 500 µL of the spore suspension from the *can1* mutant was combined with 500 µL of the spore suspension from the *fur1* mutant, for a total volume of 1 mL. The spore mixture was spread across 10 individual V8 pH 5.0 plates (100 µL per plate) using 2 mm glass beads. Plates were incubated at room temperature in the dark for 7 days, after which spores were scraped and transferred to YNB plates containing 60 µg/mL canavanine and 100 µg/mL 5-FC, following incubation at room temperature until colonies appeared (approximately 3 weeks). A single double-drug-resistant colony from each plate was purified by streaking onto YNB + canavanine + 5-FC and then transferred to V8 pH 5.0 for gDNA isolation as described above.

To determine whether the double-drug-resistant isolates were recombinant, the *CAN1* and *FUR1* genes were interrogated by PCR and restriction digest to determine whether the mutant or wild-type allele was present. In CBS7841, the *can1-2* mutant allele, selected for downstream experiments, and the wild-type *CAN1* alleles were distinguished by PCR with wild-type and mutant-specific primers (primer pairs JOHE45891/JOHE45184 and JOHE45360/JOHE45184, respectively) each pair producing a 651 bp fragment (**Figure 7A**). For CBS7855, an 861 bp region of the *CAN1* gene was first amplified with primer pair JOHE45183/JOHE45188 and then digested with SspI, which would not digest the wild-type *CAN1* allele but would cleave the mutant *can1-3* allele into two fragments (~560 and ~300 bp; **Figure 7A**). PCR and restriction enzyme digest was used to distinguish between the mutant and wild-type *FUR1* alleles in both strains. First, a 1250 bp region of the *FUR1* locus was amplified using primers JOHE45644 and JOHE45647 in wild-type and *fur1* mutant strains. In CBS7481, restriction enzyme digest with HpyCHIII would cleave this PCR product into two visible fragments that are distinct between wild-type and *fur1-1* mutant alleles (~600 and ~370 bp fragments

in the wild-type;~600 and ~425 bp in the *fur1-1* mutant; *Figure 7A*). In CBS7855, digestion of the 1250 bp fragment with PasI resulted in an undigested product for the *fur1-2* mutant allele and in two fragments for the wild-type allele (~840 and ~410 bp in size; *Figure 7A*).

PCR-RFLP analysis of some of the double-drug-resistant isolates did not match the parental mutant genotypes at both *CAN1* and *FUR1* loci. The most likely explanation for such results is a spontaneous mutation in one gene arising on the background of the other mutation; for example, a spontaneous *fur1* mutation on the CBS7855 *can1-3* background would result in a double-drug-resistant strain with a restriction pattern at the *CAN1* locus consistent with the *can1-3* strain but exhibit a restriction pattern consistent with wild-type at the *FUR1* locus. To validate this, 19 of the double-drug-resistant isolates that did not have both parental mutant alleles were Sanger-sequenced to determine what other mutation(s) on the background of the already known *can1* or *fur1* mutations could explain resistance to both drugs (*Figure 7—figure supplement 1*).

Meiotic recombination along Chr 3 in the progeny isolated from the SEC747 × SEC631 cross was determined by PCR amplification followed by Sanger sequencing to analyze whether a WT or a UV-induced mutation in genes L203_103150 (primer pairs JOHE51319/JOHE51320) and L203_103150 (primer pair JOHE50555/JOHE50556) had been inherited by the progeny.

## Scanning electron microscopy

Scanning electron microscopy was performed at the North Carolina State University Center for Electron Microscopy, Raleigh, NC, USA. Agar blocks (~0.5 cm$^3$) containing hyphae were excised and fixed in 0.1 M sodium cacodylate buffer (pH = 6.8) containing 3% glutaraldehyde at 4°C for several weeks. The agar blocks were rinsed with cold 0.1 M sodium cacodylate buffer pH 6.8 three times and then dehydrated in a graded series of ethanol to reach 100% ethanol. The blocks were subjected to critical-point drying with liquid $CO_2$ (Tousimis Research Corp.) and sputter coated with 50 Å of gold/palladium using a Hummer 6.2 sputter coater (Anatech USA). The samples were viewed at 15 kV with a JSM 5900LV scanning electron microscope (JEOL) and captured with a Digital Scan Generator (JEOL) image acquisition system.

## *C. depauperatus* confrontation assays

Using a toothpick, spores or hyphal cells were scraped from V8 pH 5.0 plates and struck onto MS plates in a straight line. A gap of approximately 2 mm was left between the two strains on the plate. Plates were incubated for 2 weeks in the dark at room temperature and photographed with a Zeiss Axiocam 105 color camera attached to a Zeiss Scope.A1 microscope.

## Analysis of basidiospores derived from co-cultures of *mfα∆* and *ste3∆* mutants

Hyphae of *mfα∆* (strain SEC831) and *ste3∆* (strain SEC836) mutants were isolated from V8 pH 5.0 plates. 50 µL of each cell suspension was spotted onto MS medium either alone or together in one spot. After two weeks, spores from single basidia were microdissected onto YPD medium, and allowed to germinate. Cells from individual basidiospores were plated onto YPD and YPD + NAT, gDNA was extracted, and the *MFα* and *STE3* genes were interrogated by PCR using the diagnostic primers listed in *Supplementary file 2* for the corresponding gene or 3' junctions of *mfα∆::NAT* and *ste3∆::NAT* deletion loci.

## Acknowledgements

We thank Vikas Yadav for critical reading of the manuscript, Fred Dietrich for computational resources, Valerie Lapham of the Center for Electron Microscopy at North Carolina State University for assistance with SEM, and the Broad Institute Genomics Platform for generating Illumina sequences. We are indebted to Rytas Vilgalys for inspiring us to study the unusual homothallic life cycle of *Cryptococcus depauperatus*. This study was supported by NIH/NIAID R01 Award AI50113-17 and NIH/NIAID R01 Award AI39115-24 awarded to JH, R01 grant AI33654-04 awarded to JH, David Tobin, and Paul Magwene, and NIH/NHGRI grant U54HG003067 to the Broad Institute. MN acknowledges support from German Research Foundation, DFG grant NO407/7-2. JH is also Co-Director and Fellow of the CIFAR program *Fungal Kingdom: Threats & Opportunities*.

## Additional information

### Funding

| Funder | Grant reference number | Author |
|---|---|---|
| National Institute of Allergy and Infectious Diseases | AI50113-17 | Joseph Heitman |
| National Institute of Allergy and Infectious Diseases | AI39115-24 | Joseph Heitman |
| National Institute of Allergy and Infectious Diseases | AI33654-04 | Joseph Heitman |
| National Institutes of Health | U54HG003067 | Christina A Cuomo |
| German Research Foundation | NO407/7-2 | Minou Nowrousian |

The funders had no role in study design, data collection and interpretation, or the decision to submit the work for publication.

### Author contributions

Andrew Ryan Passer, Conceptualization, Formal analysis, Investigation, Methodology, Resources, Writing - preliminary draft as a thesis chapter, Writing – review and editing; Shelly Applen Clancey, Conceptualization, Data curation, Investigation, Methodology, Resources, Validation, Writing – original draft, Writing – review and editing, Incorporating revisions from co-authors, Writing the final draft; Terrance Shea, Formal analysis, Validation; Márcia David-Palma, Investigation, Writing – review and editing; Anna Floyd Averette, Technical support, Writing – review and editing; Teun Boekhout, Betina M Porcel, Resources, Writing – review and editing; Minou Nowrousian, Data curation, Funding acquisition, Investigation, Methodology, Resources, Visualization, Writing – review and editing; Christina A Cuomo, Conceptualization, Data curation, Formal analysis, Funding acquisition, Methodology, Resources, Writing – review and editing; Sheng Sun, Conceptualization, Formal analysis, Investigation, Methodology, Validation, Writing – review and editing; Joseph Heitman, Conceptualization, Funding acquisition, Project administration, Supervision, Writing – original draft, Writing – review and editing; Marco A Coelho, Conceptualization, Data curation, Formal analysis, Investigation, Methodology, Resources, Validation, Visualization, Writing – original draft, Writing – review and editing, Incorporating revisions from co-authors, Writing the final draft

### Author ORCIDs

Minou Nowrousian  http://orcid.org/0000-0003-0075-6695
Christina A Cuomo  http://orcid.org/0000-0002-5778-960X
Sheng Sun  http://orcid.org/0000-0002-2895-1153
Joseph Heitman  http://orcid.org/0000-0001-6369-5995
Marco A Coelho  http://orcid.org/0000-0002-5716-0561

### Decision letter and Author response

Decision letter https://doi.org/10.7554/eLife.79114.sa1
Author response https://doi.org/10.7554/eLife.79114.sa2

## Additional files

### Supplementary files

- Supplementary file 1. Strains used in this study.
- Supplementary file 2. Primers used in this study.
- MDAR checklist

### Data availability

Sequencing reads and genome assemblies of C depauperatus CBS7841 and CBS7855 were submitted to GenBank under BioProjects PRJNA200572 and PRJNA200573, respectively. All other genomic

data (RNA-seq and Illumina sequence of C depauperatus CBS7841 can1 mutants) are available under BioProject PRJNA803141. Source data files have been provided for Figures 1 to 7.

The following datasets were generated:

| Author(s) | Year | Dataset title | Dataset URL | Database and Identifier |
|---|---|---|---|---|
| Broad Institute | 2016 | Cryptotoccus Sequencing | https://www.ncbi.nlm.nih.gov//bioproject/PRJNA200572 | NCBI BioProject, PRJNA200572 |
| Broad Institute | 2016 | Cryptotoccus Sequencing | https://www.ncbi.nlm.nih.gov/bioproject/PRJNA200573 | NCBI BioProject, PRJNA200573 |
| Passer AR, Coelho MA | 2022 | Cryptococcus depauperatus raw sequence reads | https://www.ncbi.nlm.nih.gov/bioproject/PRJNA803141 | NCBI BioProject, PRJNA803141 |

The following previously published datasets were used:

| Author(s) | Year | Dataset title | Dataset URL | Database and Identifier |
|---|---|---|---|---|
| Janbon G | 2012 | Cryptococcus neoformans var. grubii H99 genome | https://www.ncbi.nlm.nih.gov/bioproject/PRJNA411/ | NCBI BioProject, PRJNA411 |
| Cuomo CA | 2016 | Cryptotoccus Sequencing | https://www.ncbi.nlm.nih.gov/bioproject/PRJNA200571 | NCBI BioProject, PRJNA200571 |
| Cuomo CA | 2016 | Cryptotoccus Sequencing | https://www.ncbi.nlm.nih.gov/bioproject/PRJNA191370 | NCBI BioProject, PRJNA191370 |
| Passer RA, Coelho MA | 2019 | Cryptococcus floricola strain DSM 27421 Genome sequencing and assembly | https://www.ncbi.nlm.nih.gov/bioproject/PRJNA496466 | NCBI BioProject, PRJNA496466 |
| Passer RA, Coelho MA | 2019 | Cryptococcus wingfieldii strain CBS7118 Genome sequencing and assembly | https://www.ncbi.nlm.nih.gov/bioproject/PRJNA496468 | NCBI BioProject, PRJNA496468 |
| Loftus BJ | 2005 | WGS sequencing of strain JEC21 (serotype D) | https://www.ncbi.nlm.nih.gov/bioproject/PRJNA13856/ | NCBI BioProject, PRJNA13856 |
| D'Souza CA | 2011 | Cryptococcus gattii WM276 RefSeq Genome | https://www.ncbi.nlm.nih.gov/bioproject/PRJNA62089 | NCBI BioProject, PRJNA62089 |
| Cuomo CA | 2013 | Kwoniella mangroviensis CBS 8507 | https://www.ncbi.nlm.nih.gov/bioproject/PRJNA352839 | NCBI BioProject, PRJNA352839 |
| Cuomo CA | 2013 | Kwoniella mangroviensis CBS 10435 | https://www.ncbi.nlm.nih.gov/bioproject/PRJNA202099 | NCBI BioProject, PRJNA202099 |

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
