## [Editor Report]

There are various ways in which self-fertility has arisen in the fungal kingdom. This study describes a novel form of self-fertility that evolved in a species closely related to the *Cryptococcus* species causing serious human lung and brain infections, in which sexual development is achieved by self-signaling of a cognate pheromone and pheromone-receptor pair. Through a combination of high-quality genomic analysis and experimental gene expression and manipulation work, the study significantly adds to our understanding of the evolution and flexibility of fungal breeding systems.

---

## [Decision Letter]

**Decision letter after peer review:**

Thank you for submitting your article "Obligate sexual reproduction of a homothallic fungus closely related to the *Cryptococcus* pathogenic species complex" for consideration by *eLife*. Your article has been reviewed by 2 peer reviewers, and the evaluation has been overseen by a Reviewing Editor and Detlef Weigel as the Senior Editor. The following individual involved in the review of your submission has agreed to reveal their identity: Paul Dyer (Reviewer #1).

Essential revisions:

Both reviewers have made numerous suggestions on how to improve the clarity and accuracy of the presentation of your findings. Please consider them carefully and revise the manuscript accordingly.

*Reviewer #1 (Recommendations for the authors):*

The authors are congratulated on a significant and very interesting study that has been performed at a very high standard. Indeed, the fact that the study is so comprehensive and thorough has inevitably meant that there are some required corrections, clarifications, and suggestions to improve the manuscript given the very long length, as listed below (line numbering from supplied A4 PDF).

(1) Lines 94-96 and lines 812-813. Can add the example of homothallism in lichen-forming fungi as an example of reproductive assurance in species with patchy distributions, pre-dating the other citations [Murtagh et al., (2000). Nature 404: 564. https://doi.org/10.1038/35007142].

(2) Lines 105. Close up space after C. neoformans.

(3) Lines 115, 315 Need to hyphenate 'mating-type' where used as an adjective. By contrast, remove hyphens where not an adjective e.g. line 146.

(4) Line 116. For preciseness use "Among these genes are those encoding for the mating-type specific pheromones…."

(5) Line 127. Need to remove accidental 'the' after "unique to the mating type (the) of the cell".

(6) Line 252. Refer to five 'major' inversions. However, arguably only two of these are 'major' and the others involve a relatively minor number of genes. So authors might wish to rephrase this sentence?

(7) Line 264. States that nearly zero sequence divergence seen except for 'the' (my emphasis) rDNA locus. This indicates that there was only a single rDNA locus in the genome. However, normally there are multiple rDNA loci and Figure 1A Circos plot seems to confirm this? So please clarify this statement e.g. make it clear applies to only one of several rDNA loci if this is the case.

(8) Lines 301-320. General question (perhaps beyond this paper but for thought). How does the genome sequence divergence specifically at the MAT locus compare to the general sequence divergence between the two isolates of C. depauperatus? Could give clues as to whether this region exhibits restricted cross over as seen in some other fungal systems.

(9) Line 362. Use of 'however' is redundant in this sentence as no preceding contrast. So can delete it?

(10) Lines 555-561. Gene deletion is stated to be confirmed by PCR. Was Southern blotting also used as backup? Admittedly perhaps not essential now with PCR diagnostics.

(11) Line 573. Reword to "…process in C. depauperatus can occur independently of pheromone-receptor signaling…" as more nuanced than a simple "is independent of".

(12) Lines 742-749. Must admit I am confused here. From previous logic it seems that only recombinant CAN and FUR spores containing complementary recessive alleles from the mating partners should be recovered from the selective media, and yet it is stated that double-drug resistant isolates were recovered that lacked "the expected mutant parent alleles". So what was their origin (also listed in Figure 7D where only 1.9-97.3% were recombinant leaving many that were not)? Is it possible, for example, that the spore collection method also harvested heterokaryotic basidia and hyphae that had not undergone meiosis but had the complementary parent nuclei? Please explain.

(13) Lines 915-919. Studies in Aspergillus nidulans have shown that homothallism in this species does not bypass the requirements for partner signalling, but instead requires activation of the heterothallic signalling pathway for sexual development in a single individual. Could cite this work as a nice parallel here to the findings in C. depauperatus. See Paoletti et al., (2007), Current Biology 17: 1384-1389; https://doi.org/10.1016/j.cub.2007.07.012

(14) Line 923. For clarity change to “…genes, including the MF*α* gene encoding the mating pheromone precursor, are found outside of…”.

(15) Line 923. Refers to lipophilic peptides, but to clarify would the MF*α* gene product be expected to be a hydrophilic α rather than lipophilic a-type, so argument not apply to the same extent?

(16) Line 1079. Need close bracket rather than semi-colon after the web address.

(17) Methods section. The ‘RNA extraction and RNA-seq analysis” section include catalog numbers, which are missing elsewhere. So omit for consistency?

(18) Line 1246. For clarity change to “The C. depauperatus pheromone precursor encoding gene was amplified…”.

(19) Line 1247, 1256, 1279. Recommend to insert species from which the various promotor and upstream regions were amplified for clarity. Seems was C. neoformans, but in the main text, the ACT1 promotor from C. depauperatus is mentioned? And what is the ACT1 gene, is this a housekeeping high activity one – please provide some background.

(20) Line 1321. How long were spores exposed to UV, no time period mentioned?

(21) Line 1417. Which primers were used for ‘PCR interrogation’, very little detail at this very end?

*Reviewer #2 (Recommendations for the authors):*

L970: After finishing reading the manuscript, I realized that I don’t have a clear picture of what the authors envisage the life cycle of C. depauperatus to be, and what precisely they mean by the terms “obligate sexual reproduction” (title) or “constitutive sexual reproduction” (L819). As stated on L66, most fungi have a balance of sexual cycles and asexual (mitotic) cycles of cell division. I agree that the evidence in this manuscript says that sexual cycles are occurring at a high frequency in C. depauperatus, and previous work has indicated that there are no yeast-form cells. But are the authors proposing that mitotic cell divisions do not occur at all in this species [or only during the formation of chains of spores]? It’s not clear to me whether or not this is their proposal, or indeed whether there is any data pertinent to this point. When a haploid basidiospore is released from a basidium, is its only option to mate rather than to divide vegetatively? Perhaps some of these issues have been discussed in Kwon-Chung’s papers from the 1970s, but it would be good to repeat them here for a new generation of readers.

L362-365: It took me a while to realize that Ste3a is the receptor for α-pheromone and that its name derives from the fact that it is present only in MATa strains (of C. neoformans). This fact is explained later (L494-497) but it would be clearer if L362 stated that these are a pheromone and its receptor, rather than that they are “alleles of opposite mating types”.

L451: I'm not familiar with LOX analysis, but the text seems to imply that a P-value of 1 is statistically significant in this method which is counterintuitive. Some more explanation of what LOX is, and how it differs from DEseq2, would be helpful.

L556-560: The two sentences on these lines seem to contradict each other. The first seems to say that you obtained only 5 deletion mutant strains. The second seems to say that you obtained 41 deletion mutant strains.

L561-613: (1) In Figure 5B, are the orange bars simply 100% minus the blue bars? Or is there a third small group of basidia that were unclassifiable? If there is no third category, I don’t think that side-by-side bars are a clear way to present the result. A single stacked bar would be better, or simply omit the orange bars.

(2) Figure 5B includes results for a mfαΔ ste3Δ double mutant, but it doesn’t seem to be mentioned in the text.

L186: “the hypothesis was put forth…”. This statement needs to cite a reference, probably early Kwon-Chung papers.

L634: Delete “because”.

---

## [Author Response]

Reviewer #1 (Recommendations for the authors):The authors are congratulated on a significant and very interesting study that has been performed at a very high standard. Indeed, the fact that the study is so comprehensive and thorough has inevitably meant that there are some required corrections, clarifications, and suggestions to improve the manuscript given the very long length, as listed below (line numbering from supplied A4 PDF).

We thank the reviewer for the positive and constructive remarks. We have introduced several modifications to the manuscript in response to the reviewer’s concerns and suggestions.

(1) Lines 94-96 and lines 812-813. Can add the example of homothallism in lichen-forming fungi as an example of reproductive assurance in species with patchy distributions, pre-dating the other citations [Murtagh et al., (2000). Nature 404: 564. https://doi.org/10.1038/35007142].

We thank the reviewer for this suggestion and have included this appropriate reference in the revised manuscript.

(2) Lines 105. Close up space after C. neoformans.

Revised as suggested.

(3) Lines 115, 315 Need to hyphenate 'mating-type' where used as an adjective. By contrast, remove hyphens where not an adjective e.g. line 146.

We followed the reviewer recommendation and revised all of the occurrences as suggested.

(4) Line 116. For preciseness use "Among these genes are those encoding for the mating-type specific pheromones…."

Revised as suggested.

(5) Line 127. Need to remove accidental 'the' after "unique to the mating type (the) of the cell".

Thank you for pointing this out. We have removed the extra word as suggested.

(6) Line 252. Refer to five 'major' inversions. However, arguably only two of these are 'major' and the others involve a relatively minor number of genes. So authors might wish to rephrase this sentence?

We understand the reviewer’s viewpoint here and have rephrased as follows on lines 231-236: “…the chromosome structure seems to be overall conserved between the two *C. depauperatus* isolates, except for five inversions (two large and three small, of which four are coupled with duplicated sequences at the borders), and the predicted centromeric regions that differ considerably in length between some of the homologous chromosomes (figure 1—figure supplement 3).”

(7) Line 264. States that nearly zero sequence divergence seen except for 'the' (my emphasis) rDNA locus. This indicates that there was only a single rDNA locus in the genome. However, normally there are multiple rDNA loci and Figure 1A Circos plot seems to confirm this? So please clarify this statement e.g. make it clear applies to only one of several rDNA loci if this is the case.

We appreciate the reviewer’s comment. However, we found that in both *C. depauperatus* strains the rDNA repeats are located at a single locus on Chr. 1 as a tandem array. This has been observed in other species, including the budding yeast *S. cerevisiae* and e.g., in *C. neoformans*. The assembled region in each of the *C. depauperatus* strains contains six complete copies of the rDNA, which, nonetheless, should be a collapsed representation of the full-size array. An interesting aspect, however, is that the rDNA array in *C. depauperatus* is only composed of 28S-5.8S-18S genes. The 5S rRNA genes are not part of the tandemly repeated rDNA array. Instead, they are dispersed throughout the genome, as observed in fungi such as *Neurospora crassa* (Metzenberg et al., 1985), *Aspergillus nidulans* (Bartnik and Borsuk 1983) and *Schizosaccharomyces pombe* (Mao et al., 1982). Interestingly, within the *Cryptococcus* genus, there seem to be two major rDNA array configurations as shown in Author response image 1: (i) one, where each rDNA unit also comprises a copy of the 5S rRNA gene positioned in the intergenic spacer (e.g., as in *C. gattii*, *C. neoformans*, *C. deneoformans*); and another (ii) where the 5S rRNA genes are localized separately (as in *C. depauperatus*, *C. amylolentus*, etc.)

**Author response image 1. sa2fig1:** 

Interestingly, these observed differences in the rDNA cluster between *C. neoformans* and *C. depauperatus* have been previously reported by Kwon-Chung et al., (Kwon-Chung et al., 1997). We concede that the circos plot might not illustrate all of this information, but it shows that the 5S genes (marked in darker blue) are found scattered in the genome, while the 18-28S rDNA array (lighter blue) is found only on Chr. 1. As we think these details and differences are not essential for the current manuscript, we would prefer to present and discuss these and other comparative genomic analyses in a manuscript currently being prepared that will focus on the structural genomic features across the diversity of the *Cryptococcus* genus. Nevertheless, to clarify the point raised by the reviewer, we revised the circos plot key on Figure 1 by adding the 5.8S gene to the rDNA array [now reads: “18S-5.8S-28S (rDNA array)”] and modified the text between lines 241-245 so that it now reads as follows:“A sliding window analysis detected a relatively uniform pattern of sequence divergence across the genome and no evidence of introgression between the two isolates as shown by the absence of genomic tracts with nearly zero sequence divergence (except for the rDNA array, composed of 18S-5.8S-28S, which is found as a single unit on Chr 1)…”

(8) Lines 301-320. General question (perhaps beyond this paper but for thought). How does the genome sequence divergence specifically at the MAT locus compare to the general sequence divergence between the two isolates of C. depauperatus? Could give clues as to whether this region exhibits restricted cross over as seen in some other fungal systems.

We thank the reviewer for this important question. It is usually the case that the divergence within the *MAT* locus is higher between opposite mating types of the same species compared to other genomic regions, and this is usually a signature of suppression of recombination. Within basidiomycetes, there are now multiple documented examples in which recombination suppression evolved repeatedly, linking the two mating-type loci (*P/R* and *HD*), followed by subsequent extension of the region beyond the mating type genes. One such example is the *Cryptococcus* pathogenic clade (such as in *C. neoformans* and *C. gattii*) discussed in this manuscript, in which the *MAT* locus is highly rearranged between opposite mating types and suppressed for recombination via crossing-over [although recombination via gene conversion can still occur (Sun et al., 2012)], resulting in a higher divergence at *MAT* compared to other genomic regions.

In *C. depauperatus*, we considered that it would be less likely to see an elevated divergence at the proposed *MAT* locus given homothallism, and the fact that in *C. depauperatus*, the haploid nuclei that fuse prior to meiosis are completely identical in intra-strain crosses and this region is also structurally similar between the two strains. Thus, recombination with crossing-over is likely to occur within the *MAT* locus region of *C. depauperatus*, similar to that observed during α-α unisexual reproduction in *C. deneoformans* (Sun et al., 2014). We also mention that the proposed *MAT* locus in *C. depauperatus* underwent substantial remodeling via genome reshuffling and gene loss compared to both *C. amylolentus* and *C. neoformans MAT* loci, which itself likely indicates that in *C. depauperatus* there is minimal or no selection maintaining this region as a cluster of tightly linked loci, in contrast to that observed in heterothallic species with defined mating types.

Despite this reasoning, we followed the reviewer suggestion, and measured the divergence between the two *C. depauperatus* strains for the proposed *MAT* locus region using the approach previously described in the Material and Methods. We found that the average divergence for the *MAT*-containing chromosome (Chr 4) is ~1.6% (shown in Figure 1—figure supplement 4), while the divergence falls to ~1.3% within the *MAT* region (*k*, with Jukes-Cantor correction). Comparable values were also obtained when using Dnadiff, which is a wrapper for the NUCmer alignment program from MUMmer, that quantifies the differences between two sequences. Therefore, we think this level of divergence within *MAT* is unlikely to be associated with any form of suppression for recombination.

(9) Line 362. Use of 'however' is redundant in this sentence as no preceding contrast. So can delete it?

Thank you for pointing this out. We revised as recommended and expanded to accommodate a suggestion from reviewer #2 (lines 309-313):

“Sequence alignments and phylogenetic analyses further showed that the predicted product of the MF gene is highly similar to other α/A1 pheromones (Figure 2B), whereas Ste3 clusters together with α/A2 alleles from other Cryptococcus species (Figure 2C), indicating they might constitute a compatible pheromone-receptor pair.”

(10) Lines 555-561. Gene deletion is stated to be confirmed by PCR. Was Southern blotting also used as backup? Admittedly perhaps not essential now with PCR diagnostics.

We admittedly did not confirm gene deletions by Southern blotting. All deletions were instead carefully inspected by in-gene, spanning, and junction (5’ and 3’ flanking) PCRs, which we think provide equivalent validation. To make it more transparent, all the gels are now provided as source information associated with the corresponding figures. Specifically, Figure 4–source data 1, related to ectopic expression of *C. depauperatus MF*a gene in *C. neoformans* WT, *ste3* and *ste6* deletion mutants; and Figure 5—figure supplement 1–source data 1, related with the construction of *mf*a, *ste3* and *dmc1* deletion mutants in *C. depauperatus*.

(11) Line 573. Reword to "…process in C. depauperatus can occur independently of pheromone-receptor signaling…" as more nuanced than a simple "is independent of".

Agreed and revised as suggested.

(12) Lines 742-749. Must admit I am confused here. From previous logic it seems that only recombinant CAN and FUR spores containing complementary recessive alleles from the mating partners should be recovered from the selective media, and yet it is stated that double-drug resistant isolates were recovered that lacked "the expected mutant parent alleles". So what was their origin (also listed in Figure 7D where only 1.9-97.3% were recombinant leaving many that were not)? Is it possible, for example, that the spore collection method also harvested heterokaryotic basidia and hyphae that had not undergone meiosis but had the complementary parent nuclei? Please explain.

We were also initially intrigued by the fact that we were recovering double-drug resistant progeny that did not have either one or the other can1 or fur1 parental mutant alleles. However, upon inspection of some of these isolates by PCR amplification and Sanger sequencing (results given in Figure 7—figure supplement 1) we uncovered that some of the double-drug resistant progeny had gained de novo spontaneous mutations in *CAN1* or *FUR1* loci that were different from those of the parental isolates. This information was previously stated (now in lines 576-582) as follows:

“When PCR-RFLP analysis did not match the parental genotypes at both loci, the isolate was not scored as a progeny despite being double-drug resistant. Rather than cell fusion and meiosis generating the double can1 fur1 mutant resistant to both canavanine and 5-FC, Sanger sequencing revealed that in such cases a spontaneous mutation had arisen on the background of the already known can1 or fur1 parental mutations (Figure 7—figure supplement 1). Therefore, only resistant isolates that contained the parental *can1* and *fur1* mutant alleles upon validation by PCR-RFLP were scored as a recombinant progeny.”

We further elaborate in the Material and Methods section (lines 1188-1197) about the origin of these non-recombinant, double-drug resistant, isolates and the approach used to score only the recombinant ones.

We also think the possibility of having harvested heterokaryotic basidia or hyphae that had not undergone meiosis instead of meiotic products is very unlikely, particularly because as we note in lines 564-567:

“these mutations both confer recessive drug resistance; thus, in crosses of can1 and fur1 mutants, this allows haploid meiotic F1 progeny to be selected by virtue of resistance to both drugs, whereas any diploid fusion products (*can1*/*CAN1 FUR1*/*fur1*) would be sensitive to both drugs as a result of complementation.”

(13) Lines 915-919. Studies in Aspergillus nidulans have shown that homothallism in this species does not bypass the requirements for partner signalling, but instead requires activation of the heterothallic signalling pathway for sexual development in a single individual. Could cite this work as a nice parallel here to the findings in C. depauperatus. See Paoletti et al., (2007), Current Biology 17: 1384-1389; https://doi.org/10.1016/j.cub.2007.07.012

We thank the reviewer for this excellent suggestion. It is indeed a nice parallel to the findings in *C. depauperatus* and, thus, we expanded this section to include this interesting study (lines 730-734).

(14) Line 923. For clarity change to "…genes, including the MFα gene encoding the mating pheromone precursor, are found outside of…".

Clarified as suggested.

(15) Line 923. Refers to lipophilic peptides, but to clarify would the MFα gene product be expected to be a hydrophilic α rather than lipophilic a-type, so argument not apply to the same extent?

We appreciate the reviewer’s feedback. However, while in ascomycetes, the pheromones for the two mating types are asymmetric with respect to size and hydrophobicity (the α-factor is an unmodified peptide, and the **a**-factor is farnesylated and carboxymethylated at a C-terminal CAAX box, making it very hydrophobic), this asymmetry is not observed in basidiomycetes. Instead, all basidiomycetes, irrespective of their mating type, express lipid-modified, **a**-factor-like, pheromones, and “chemo-sensing” specificity is mediated by allelic variants of the **a**-factor/Ste3 genes (designated, in its simplest configuration in opposite mating types, as *MF*α/*STE3*α and *MF***a**/*STE3***a**). Therefore, as both mating types produce poorly diffusible pheromones, duplication of these genes in some basidiomycetes may have been selected for to promote more efficient foraging for mating partners. In contrast, the presence of a single *MF*α gene in *C. depauperatus* suggests that selection may have acted to reduce the energetic burden of additional gene copies without an obvious benefit, since the pheromone (*MF*α) and receptor (*STE3***a**) expressed in the same cell are intercompatible.

(16) Line 1079. Need close bracket rather than semi-colon after the web address.

Altered as suggested.

(17) Methods section. The 'RNA extraction and RNA-seq analysis" section include catalog numbers, which are missing elsewhere. So omit for consistency?

We have removed the catalog numbers from the main text following the reviewer recommendation.

(18) Line 1246. For clarity change to "The C. depauperatus pheromone precursor encoding gene was amplified…".

Revised as suggested.

(19) Line 1247, 1256, 1279. Recommend to insert species from which the various promotor and upstream regions were amplified for clarity. Seems was C. neoformans, but in the main text, the ACT1 promotor from C. depauperatus is mentioned? And what is the ACT1 gene, is this a housekeeping high activity one – please provide some background.

Thank you for this suggestion and we agree this was not completely clear. We have revised the corresponding sections in the Material and Methods and provided a few additional details to make it more transparent. The *ACT1* gene encodes for actin, and the promoter of this gene if efficient in regulating constitutive gene expression. The *C. neoformans* H99 *ACT1* promoter and terminator were employed for heterologous expression of the *C. depauperatus* pheromone gene. Plasmids for ectopic integration and gene deletion in *C. depauperatus* harbored a *NAT* resistant gene under control of the *C. depauperatus ACT1* promotor and the terminator of the phosphoribosyl anthranilate isomerase (*TRP1*) gene.

(20) Line 1321. How long were spores exposed to UV, no time period mentioned?

We thank the reviewer for raising this issue. The reason why we don’t mention a time period is because we used the “Energy Mode” selection on the Stratalinker. In this mode, we are specifying the overall amount of UV energy that is delivered to the cells rather than a specific length of exposure. However, after careful revision of the procedure employed, we noted there was a typo in the UV energy values reported originally. Instead of 100, 200 and 300 µJ/cm^2^, it should state 10,000, 20,000, and 30,000 µJ/cm^2^. This typo was due to labeling of the samples sent for sequencing in the abbreviated form that appears in the Stratalinker display (i.e., the numbers on the display represent µJ/cm^2^ × 100; e.g., setting to 100 means 100 x 100 = 10,000 µJ/cm^2^), which was then transposed into the manuscript. We apologize for this oversight, and have corrected the values in line 1132.

(21) Line 1417. Which primers were used for 'PCR interrogation', very little detail at this very end?

We appreciate this was not described with sufficient detail. We have now rephased lines 1227-1230 to: “Cells from individual basidiospores were plated onto YPD and YPD+NAT, gDNA was extracted, and the *MF*α and *STE3* genes were interrogated by PCR using the diagnostic primers listed in Supplementary File 2 for the corresponding gene or 3’ junctions of *mf*αΔ::*NAT* and *ste3*Δ::*NAT* deletion loci”.

We also altered a few details in the Supplementary File 2 to provide a better description of each of the primers used in this study.

Reviewer #2 (Recommendations for the authors):L970: After finishing reading the manuscript, I realized that I don't have a clear picture of what the authors envisage the life cycle of C. depauperatus to be, and what precisely they mean by the terms "obligate sexual reproduction" (title) or "constitutive sexual reproduction" (L819). As stated on L66, most fungi have a balance of sexual cycles and asexual (mitotic) cycles of cell division. I agree that the evidence in this manuscript says that sexual cycles are occurring at a high frequency in C. depauperatus, and previous work has indicated that there are no yeast-form cells. But are the authors proposing that mitotic cell divisions do not occur at all in this species [or only during the formation of chains of spores]? It's not clear to me whether or not this is their proposal, or indeed whether there is any data pertinent to this point. When a haploid basidiospore is released from a basidium, is its only option to mate rather than to divide vegetatively? Perhaps some of these issues have been discussed in Kwon-Chung's papers from the 1970s, but it would be good to repeat them here for a new generation of readers.

We appreciate the reviewer’s concern about the use of the terms “obligate sexual reproduction” and/or “constitutive sexual reproduction”. Unlike other *Cryptococcus* species, *C. depauperatus* does not have an asexual yeast stage, and grows exclusively as hyphae. When a spore germinates (approx. after 48h), it produces monokaryotic hyphae extending from the two poles of the germinating spore and the nuclei in these hyphae divide by mitosis. Hyphae maturation and basidia formation occurs soon after (~144-196 h) followed by meiosis and repeated rounds of mitosis to produce long chains of spores. Therefore, *C. depauperatus*, seems to be permanently undergoing sexual reproduction. In other words, only the sexual developmental program seems to be active at any given time during the life cycle. While this may not be exactly the same as obligate sexual reproduction in animals, it does mean that every spore that is produced results from meiotic event and there are no asexual structures/propagules (e.g., conidia). Among other fungi, the homothallic ascomycete *Sordaria macrospora* has also been considered obligately sexual (Teichert et al., 2014), and most basidiomycete mushroom-forming species of the Agaricomycotina subphylum are considered to be so as well.

To clarify this, we rephrased a statement in lines 186-189 such that now read as:

“Considering the striking resemblance of its growth with the sexual life cycle of other *Cryptococcus* species, the hypothesis was put forth that *C. depauperatus* is homothallic and only the sexual developmental program is active at any given time during the life cycle of this species (Malloch et al., 1978; Kwon-Chung et al., 1995; Rodriguez-Carres et al., 2010).”

We also replaced the term “constitutive” or ”constitutively” by terms such as “continuously” (e.g., lines 639, 795) when referring to the *C. depauperatus* sexual cycle.

L362-365: It took me a while to realize that Ste3a is the receptor for α-pheromone and that its name derives from the fact that it is present only in MATa strains (of C. neoformans). This fact is explained later (L494-497) but it would be clearer if L362 stated that these are a pheromone and its receptor, rather than that they are "alleles of opposite mating types".

We have updated the revised manuscript as recommended such that this section now reads as (lines 309-313):

“Sequence alignments and phylogenetic analyses further showed that the predicted product of the *MF* gene is highly similar to other α/A1 pheromones (Figure 2B), whereas Ste3 clusters together with a/A2 alleles from other Cryptococcus species (Figure 2C), indicating they might be a compatible pheromone-receptor pair”.

We hope this change provides enough clarification at this point in the manuscript. Later, we provide further experimental evidence confirming this initial result obtained from sequence/phylogenetic analyses.

L451: I'm not familiar with LOX analysis, but the text seems to imply that a P-value of 1 is statistically significant in this method which is counterintuitive. Some more explanation of what LOX is, and how it differs from DEseq2, would be helpful.

We appreciate the reviewer’s concern and have provided a few additional details in the Material and Methods section (lines 1042-1046). LOX (Levels Of eXpression) reports (among other statistics) Bayesian p-values for differential expression (essentially, likelihoods that a gene is differentially expressed), in contrast to methods like DESeq, which report p-values that indicate statistical significance (essentially, likelihoods that a gene is not differentially expressed). Therefore, with LOX, a larger p-value is better (maximum would be 1, and we have found LOX to be relatively "optimistic" in its assessment, so anything below 1 should be taken as most likely not differentially expressed in the dataset). The reason why we used LOX in addition to DESeq2 is because LOX uses a different type of underlying statistics that is not sensitive to the percentage of differentially expressed genes (in contrast to DESeq2, which might give unreliable results if too many genes are differentially expressed, because it works with the underlying assumption that the majority of genes are not differentially expressed). Therefore, if both methods (LOX and DESeq2) indicate that a gene is differentially expressed, one can be sure that this is not biased by some specific underlying statistical method.

L556-560: The two sentences on these lines seem to contradict each other. The first seems to say that you obtained only 5 deletion mutant strains. The second seems to say that you obtained 41 deletion mutant strains.

We thank the reviewer for this observation and agree that the two sentences as stated could be misleading. Therefore, we decided to remove second sentence (line 472-474) which described to the average proportion of analyzed transformants obtained from a given transformation experiment that had a correct gene deletion.

L561-613: (1) In Figure 5B, are the orange bars simply 100% minus the blue bars? Or is there a third small group of basidia that were unclassifiable? If there is no third category, I don't think that side-by-side bars are a clear way to present the result. A single stacked bar would be better, or simply omit the orange bars.

The reviewer is correct! These represent percentage of basidia with and without spore chains. As suggested, we changed the graph on Figure 5B to a stacked bar plot to better illustrate the data. For consistency we similarly changed the layout of the graph on Figure 6B. The corresponding figure legends were revised where needed to accommodate these changes. Also, for simplicity of presentation, we removed the individual counts (represented as white dots in the previous version of the two graphs), but all the numerical raw data is given as source data (Figure 5–source data 1 and Figure 6–source data 1).

(2) Figure 5B includes results for a mfαΔ ste3Δ double mutant, but it doesn't seem to be mentioned in the text.

Thanks for raising this point. The *mf*αΔ *ste3*Δ double mutant is mentioned the legend of Figure 5 and was isolated from progeny of mfαΔ x ste3Δ co-cultures. We concur that this should also appear in the text and have introduced a new sentence in lines 487-489 that states:

“A *mf*αΔ *ste3*Δ double mutant isolated from progeny of *mf*αΔ x *ste3*Δ co-cultures (see next section) displayed similar defects as the single mutants (Figure 5B).”

L186: "the hypothesis was put forth…". This statement needs to cite a reference, probably early Kwon-Chung papers.

We cited a previous paper from the lab and two other references (Malloch et al., 1978; Kwon-Chung et al., 1995; Rodriguez-Carres et al., 2010) that we think are the most suitable to support the revised statement in lines 186-189.

L634: Delete "because".

Thank you for pointing this out. We have removed the extra word as suggested.

References

Bartnik E, Borsuk P. 1983. Transcription of the rRNA gene cluster in Aspergillus nidulans. Curr Genet 7: 113-115.

Guterres DC, Ndacnou MK, Saavedra-Tobar LM, Salcedo-Sarmiento S, Colman AA, Evans HC, Barreto RW. 2021. Cryptococcus depauperatus, a close relative of the human-pathogen C. neoformans, associated with coffee leaf rust (Hemileia vastatrix) in Cameroon. Braz J Microbiol 52: 2205-2214.

Kwon-Chung KJ, Chang YC, Bauer R, Swan EC, Taylor JW. 1995. The characteristics that differentiate Filobasidiella depauperata from Filobasidiella neoformans. Stud Mycol 38: 67-79.

Kwon-Chung KJ, Chang YC, Penoyer L. 1997. Species of the genus Filobasidiella differ in the organization of their 5S rRNA genes. Mycologia 89: 244-249.

Malloch D, Kane J, Lahai DG. 1978. Filobasidiella arachnophila sp. nov. Can J Bot 56: 1823-1826.

Mao J, Appel B, Schaack J, Sharp S, Yamada H, Söll D. 1982. The 5S RNA genes of *Schizosaccharomyces pombe*. Nucleic Acids Res 10: 487-500.

Metzenberg RL, Stevens JN, Selker EU, Morzycka-Wroblewska E. 1985. Identification and chromosomal distribution of 5S rRNA genes in *Neurospora crassa*. Proc Natl Acad Sci U S A 82: 2067-2071.

Rodriguez-Carres M, Findley K, Sun S, Dietrich FS, Heitman J. 2010. Morphological and genomic characterization of Filobasidiella depauperata: a homothallic sibling species of the pathogenic Cryptococcus species complex. PLoS One 5: e9620.

Savelkoul E, Toll C, Benassi N, Logsdon JM. 2019. Multiple separate cases of pseudogenized meiosis genes Msh4 and Msh5 in Eurotiomycete fungi: associations with Zip3 sequence evolution and homothallism, but not Pch2 losses. bioRxiv doi:10.1101/750497: 750497.

Sun S, Billmyre RB, Mieczkowski PA, Heitman J. 2014. Unisexual reproduction drives meiotic recombination and phenotypic and karyotypic plasticity in Cryptococcus neoformans. PLoS Genet 10: e1004849.

Sun S, Hsueh YP, Heitman J. 2012. Gene conversion occurs within the mating-type locus of Cryptococcus neoformans during sexual reproduction. PLoS Genet 8: e1002810.

Teichert I, Nowrousian M, Poggeler S, Kuck U. 2014. The filamentous fungus Sordaria macrospora as a genetic model to study fruiting body development. Adv Genet 87: 199-244.